# A New Chengjiang Worm Sheds Light on the Radiation and Disparity in Early Priapulida

**DOI:** 10.3390/biology12091242

**Published:** 2023-09-15

**Authors:** Deng Wang, Jean Vannier, Jie Sun, Chiyang Yu, Jian Han

**Affiliations:** 1State Key Laboratory of Continental Dynamics, Shaanxi Key Laboratory of Early Life & Environments and Department of Geology, Northwest University, Xi’an 710069, China; wangdeng@nwu.edu.cn (D.W.); kekeluckly3@163.com (J.S.); yuchiyang@stumail.nwu.edu.cn (C.Y.); 2Yunnan Key Laboratory for Palaeobiology, Yunnan University, Kunming 650091, China; 3Laboratoire de Géologie de Lyon: Terre, Planètes, Environnement (CNRS-UMR 5276), CNRS, ENS de Lyon, Université Claude Bernard Lyon 1, Université de Lyon, Villeurbanne 69622, France; jean.vannier@univ-lyon1.fr; 4School of Earth Science and Resources, Key Laboratory of Western China’s Mineral Resources and Geological Engineering, Ministry of Education, Chang’an University, Xi’an 710054, China

**Keywords:** Priapulida, body plan, symmetry pattern, Chengjiang biota, early Cambrian

## Abstract

**Simple Summary:**

Priapulida form a small relict group of marine invertebrates characterized by a vermiform shape, an annulated trunk, and an eversible anterior part. Worms with a comparable body plan were diverse and abundant throughout the Cambrian Era, although uncertainties remain concerning their relation to modern priapulids. The problem lies in the lack of morphological detail (ornament, symmetry) available from Cambrian worms and the fact that a comprehensive phylogeny of Cambrian worms with a robust homology framework is not available to define priapulids. The exceptionally preserved worm *Ercaivermis sparios*, described here from the early Cambrian of China, displays an unusual octagonal symmetry, suggesting that different symmetry types may have co-existed in the early history of Priapulida, before five-fold symmetry was naturally selected to become overwhelmingly dominant.

**Abstract:**

The vast majority of early Paleozoic ecdysozoan worms are often resolved as stem-group Priapulida based on resemblances with the rare modern representatives of the group, such as the structure of the introvert and the number and distribution of scalids (a spiny cuticular outgrowth) and pharyngeal teeth. In Priapulida, both scalids and teeth create symmetry patterns, and three major diagnostic features are generally used to define the group: 25 longitudinal rows of scalids (five-fold symmetry), 8 scalids around the first introvert circle and the pentagonal arrangement of pharyngeal teeth. Here we describe *Ercaivermis sparios* gen. et sp. nov., a new priapulid from the early Cambrian Chengjiang Lagerstätte, characterized by an annulated trunk lacking a sclerotized ornament, four pairs of anal hooks and 16 longitudinal rows of scalids along its introvert and eight scalids around each introvert circle, giving the animal an unusual octoradial symmetry. Cladistic analyses resolve *Ercaivermis* as a stem-group priapulid. *Ercaivermis* also suggests that several biradial symmetry patterns (e.g., pentagonal, octagonal) expressed in the cuticular ornament, may have co-existed among early Cambrian priapulids and that the pentaradial mode may have become rapidly dominant during the course of evolution, possibly via the standardization of patterning, i.e., the natural selection of one symmetry type over others.

## 1. Introduction

Scalidophorans morphologically form a clade of ecdysozoan worms, with a relatively small number of extant species (ca. 260 species) distributed into three distinct phyla, the Kinorhyncha, Loricifera and Priapulida [1,2]. The relation of early Paleozoic scalidophorans to extant lineages of the group is not strongly established [3,4,5]. The majority of them fall into the Paleoscolecida, an extinct group of strongly ornamented worms [4,6,7], whereas others are considered as more or less closely related to Priapulida [8,9,10,11], Loricifera [12,13], Kinorhyncha [14,15] and possibly Nematomorpha [16]. These scalidophorans share key morphological traits such as an annulated trunk and more importantly an eversible introvert divided into three parts, each bearing specific and regularly distributed cuticular ornaments [8,10,11]. Caudal appendages also occur in some species [17,18]. Superficial resemblances with extant priapulids [8,17,18,19,20] and phylogenetic analyses [4,5,21,22] have led authors to assign these ancient worms to stem- or crown-group Priapulida. Most authors define Priapulida by the following characters: (1) 25 longitudinal rows of scalids, (2) eight scalids around the first introvert circle, and (3) pharynx armed with teeth in a pentagonal arrangement [8,23,24,25,26]. However, a crown-group Priapulida is defined by phylogenetic relationships [4,5,22] and not by a suite of morphological characteristics.

Compression and loss of characters are both responsible for an important lack of information on the detailed anatomy (e.g., the exact number and arrangement of scalids and pharyngeal teeth) of these Cambrian worms [16,27]. In this context, it is particularly difficult to check whether they had 25 longitudinal rows of scalids or not, and how these scalids distributed in circles (e.g., (8 + 8 + 9) in 1st-to-3rd circles [26]; the pattern (8 + 8 + 9) is repeated in all following circles).

Untypical morphologies are also worth being noted. For example, *Sicyophorus* a Cambrian worm with a loricate trunk, is resolved as stem-group Priapulida [4,5,21], even though it bears 25 longitudinal rows of scalids and 8 scalids around the first circle [28]. Even more problematic is *Priapulites* that bears only 20 longitudinal rows of scalids, but despite this was confidently assigned to the crown-group Priapulida [4,5,29,30]. This placement seems to have been influenced by the presence of caudal appendages and a five-fold symmetry arrangement of scalid rows closely resembling those of extant priapulids [8,21]. There are numerous other inconsistencies in the diagnosis of crown-group Priapulida. For example, the assumption that “8 scalids around the first introvert circle” is a symplesiomorphy of the ground pattern of the Priapulida (see larval stages, [26]), is clearly at odds with fossil data. This characteristic is also found in *Markuelia*, a Cambrian worm represented by late embryonic stages and currently assigned to the stem-group Scalidophora [3,21] or stem-group Priapulida [31]. More importantly, eight scalids around the first introvert circle are found in other scalidophoran groups such as Loricifera [32], which means that this character is not specific to Priapulida. The structure of the eversible pharynx and especially the pentagonal distribution of pharyngeal teeth into successive circles is another key diagnostic feature of crown-group Priapulida [8,23]. However, as for that of scalids, it is often strongly affected by post-mortem compression and pharynx inversion [16].

We used here micro-CT to study *Ercaivermis sparios* gen. et. sp. nov., a new worm from the Chengjiang Lagerstätte, that bears an unusual pattern of 16 longitudinal rows of scalids (i.e., eight-fold symmetry) and 8 scalids around the first introvert circle. This new species questions the morphological disparity of early Cambrian priapulids and more specifically, the diversity of symmetry modes in the group in the early stages of their evolution. We also discuss the validity of the diagnostic features usually used to define Priapulida.

## 2. Materials and Methods

### 2.1. Materials and Preservation

The fossil material comes from the Ercaicun section of the Chengjiang Lagerstätte, Yunnan Province, South China (Yu’anshan Formation, equivalent to the Cambrian Series 2, Stage 3). The Chengjiang localities have yielded a wealth of exceptionally preserved fossils and key information on early Cambrian animal life, since their discovery in the 1980s [8,9,33]. Although largely dominated by panarthropods, the Chengjiang fauna contains abundant and diverse scalidophoran worms [17,18,19,21] that resemble modern priapulid worms in their general morphology. A single specimen is described here and consists of its part (ELIEC-00312A) and counterpart (ELIEC-00312B). It lies almost parallel to the bedding plane, except for the posterior part of the trunk (see counterpart), which exhibits a slightly tilted position as revealed by micro-CT images (Figure 1 and Figure 2). ELIEC-00312 is strongly pyritized (Figure 1A,I). The darkest areas of the fossil correspond to an enriched concentration of iron, cobalt and chromium (Figure 3D–F). Phosphorus and sulfur are present in the remaining areas (Figure 2G,H). Pyritization has successfully replicated the overall 3D-shape of the worm and fine details of its external cuticular ornaments (e.g., distribution of scalids). The upper side of the specimen preserves the details of the scalids, including their shape, number, and spacing (Figure 1B–D and Figure 2A,B). In contrast, the opposite side appears more compressed along the anteroposterior axis, resulting in poorly preserved and more sparsely distributed scalids (Figure 1G and Figure 2C). The cuticle of the introvert is locally slightly folded (e.g., left edge of the upper side; Figure 1E and Figure 2C).

### 2.2. Imaging

Light photographs of the specimen were captured using a Canon EOS 5DS R (Northwest University, Xi’an, China). Microscopy computed tomography (micro-CT, Zeiss X radia 520, same institution) was performed with a pixel size of 6.5 µm for ELI-00312A and 5.8 µm for ELI-00312B (tiff-images). The imaging process utilized an accelerating voltage of 80 kV and a current of 88 µA. The micro-CT data (see Appendix A) were processed using ‘Dragonfly 4.0’ software. All figures were prepared with Photoshop CS9.

### 2.3. Element Mapping

The specimen was analyzed using an M4 Tornado micro X-ray fluorescence spectrometer (μ-XRF; Northwest University, Xi’an, China).

### 2.4. Measurement

We measured the size of the scalids and the distance between adjacent scalids using the tpsDig v.23 software from the SEM images.

### 2.5. Phylogenetic Analysis

We used the dataset of Shi et al. (ref. [31]) (96 taxa and 180 characters) and slightly modified the matrix (see Appendix A). A new character was added (number 180= “eight elements encircling the circumoral ring (the first circle) in Zone I”: (0) absent, (1) present), and new coding to character 40 “Number of elements comprising the first three rings and, hence, defining the number of longitudinal rows of elements in Zone I (assuming there are more than three”: (3) = 30) and character 147 “Number of terminally posterior spines, hooks”: (5) = 4 pairs). Parsimonious analyses were performed with TNT v.1.5 using New Technology Search (Driven Search with Sectorial Search, Ratchet, Drift, and Tree fusing options activated) in standard settings under equal and implied weights [34,35]. The analysis was set to find the minimum tree length 100 times and to collapse trees after each search, and all characters were treated as unordered. Repetitions with variable concavity values (k) were used to explore the effect of different degrees of homoplasy penalization to test the robustness of the dataset [36]. Probabilistic tree searches used the MK model for discrete morphological character data [37]. The maximum likelihood implementation was conducted in IQ-Tree [38], with nodal support assessed by 1000 Ultrafastbootstrap (UFBoot) replicates [39,40]. Bayesian searches (MrBayes v.3.2.6) used an Mkv+Γ model [37] with 4 runs each with 4 chains for 6,000,000 generations and burn-in at 25%, which was enough to reach convergence in each case. The convergence of chains was checked by effective sample size (ESS) values over 1000 in Tracer v.1.7 [41] and 1.0 for potential scale reduction factor (PSRF) [42].

### 2.6. Terminology

We used that of (ref. [8]) to describe the introvert of *Ercaivermis* (Zones I to III). Zone I represents the region where scalids distribute in longitudinal rows parallel to the anteroposterior body axis and circles perpendicular to it. Zone III corresponds to the pharynx that bears teeth. Zone II is situated between Zone I and Zone III and generally lacks cuticular ornament.

## 3. Results

### Systematic Paleontology

Phylum PRIAPULIDA Delage et Hérouard, 1897

Genus *Ercaivermis* nov.

LSID: urn:lsid:zoobank.org:act:FB42B18B-24EC-4A4A-AA36-A490A6199DD8

Type species. *Ercaivermis sparios* sp. nov.

Diagnosis. Vermiform body subdivided into introvert and annulated trunk. Introvert armed with scalids in longitudinal rows and circles. Each circle contains eight scalids. Quincunx pattern of scalids appears on the second to ninth circles resulting in sixteen longitudinal rows in Zone I. The eight scalids around the first circle deviate from the alignment of the 16 longitudinal rows. Zone II unarmed and tapering anteriorly. Inverted pharynx with densely distributed tiny teeth decreasing in size posteriorly (exact number unclear). Four pairs of bilaterally arranged bicaudal hooks around the posterior end. Gut straight, running from middle trunk to anus.

Etymology. From Ercaicun, the locality where the specimen was found and *vermis* (Latin) meaning worm.

Remarks. *Ercaivermis* resembles *Xiaoheiqingella*, *Yunnanpriapulus*, and *Paratubiluchus* in gross morphology [17,19]. The introvert of *Ercaivermis* is tapering anteriorly, whereas that of the latter three is swollen. Papillae distributed in rings, seen in *Xiaoheiqingella* and *Yunnanpriapulus* are not found in *Ercaivermis*. *Ercaivermis* bears four pairs of caudal hooks around the anal region, thus contrasting with the bursa, short projection or caudal appendages of *Eximipriapulus*, *Yunnanpriapulus,* and *Xiaoheiqingella*, respectively. A new genus, *Ercaivermis* is erected based on these major features.

*Ercaivermis sparios* sp. nov.

(Figure 1, Figure 2, Figure 3 and Figure 4)

LSID: urn:lsid:zoobank.org:act:87FE7750-5073-4FF4-BABB-FBD57996D1C3

Type material. Holotype ELI-00312, part and counterpart.

Locality and horizon. Yu’anshan Formation (equivalent of Cambrian Series 2 Stage 3), *Eoredlichia-Wudingaspis* zone, Chengjiang Lagerstätte, Yunnan Province, China.

Etymology. From σπάνιος (*sparios*; rare), alluding to the relative rarity of the species.

Diagnosis as for the genus.

Descriptions and comparisons. The worm is about 17 mm long (13 mm lying parallel to bedding, the remaining 4 mm being tilted and buried in the matrix (Figure 1A,I)). The body consists of an introvert and a trunk. The introvert displays three distinct zones (Zone I, II, III) from the posterior to anterior end (Figure 1A–I,G), respectively. Zone I is about 0.8 mm in longitudinal length and 2 mm in maximal width and bears spinose scalids arranged in 16 discrete longitudinal rows and 9 circles (Figure 1B–H and Figure 2). Scalids show an octagonal distribution along planes perpendicular to the anteroposterior axis of the animal (see polar-coordinate diagram, Figure 1H). Scalids of the second to ninth circles show a quincunx pattern (see white and black dots in Figure 1D,G,H and Figure 2). Notably, the eight scalids of the first introvert circle deviate from the alignment of the 16 longitudinal rows (see blue and pink dots in Figure 1D,G,H). The size of the scalids on the upper side of the specimen shows a slight increase towards the posterior end. Fewer scalids are present on the opposite site of the compressed specimen due to less favorable preservation (Figure 1G and Figure 2C).

Zone II, ca. 0.5 mm long and 1.3 mm wide, is unarmed and tapers anteriorly (Figure 1 and Figure 2). A cluster of tiny teeth is observed in the central area of Zone II (Figure 1B,D). They do not belong to the external ornament of Zone II and more likely result from the imprints of pharyngeal teeth during compression. Zone III is cylindrical and corresponds to the pharynx. It is armed with densely distributed teeth arranged in quincunx (Figure 1C,D,G and Figure 3A).

The trunk is about 15.7 mm long and has seven annulations per mm (Figure 1A–C). Its width ranges from ca. 3 mm anteriorly to 1.2 mm near the posterior end. The circumanal region is characterized by four pairs of bilaterally arranged hooks (Figure 1J,K). The gut is straight, has a central position and an evenly tubular shape. Its wall is strongly wrinkled (Figure 3B,C and Figure 4).

## 4. Discussion

### 4.1. Phylogenetic Position of Ercaivermis among Priapulids

*Ercaivermis* has an unusual distribution of scalids (16 longitudinal rows along its trunk; octoradial symmetry) that stands out from the prevailing pentaradial symmetry of most extinct and extant priapulids (typically 20 and 25 longitudinal rows of scalids [8,26]). *Ercaivermis* also has a total of 24 scalids in the first three circles (8 + 8 + 8 patten) instead of 25 (8 + 9 + 8 patten) in extant priapulids. These morphological and symmetry differences are consistent with the resolution of *Ercaivermis* as a stem-group priapulid relatively close to the crown-group (see Figure 5A and Figure A3; maximum likelihood analysis with high bootstrap support values (over 50) of key nodes).

Similar topologies are obtained via the parsimony analysis, when implied weight (k) is greater than five (see phylogenetic analyses, implied weight (k ≥ 5); Figure 5B and Figure A2B). However, when implied weight (k) is lower or equal to three and under equal weight (Figure A1 and Figure A2A), *Ercaivermis* is resolved as a crown-group Priapulida. The dataset used here and largely inherited from Shi et al. (2021) [31] was reanalyzed by Smith and Dhungana (2022). The authors noted that handling inapplicable characters (parsimony analysis [43]) resulted in polychotomy and therefore a lack of phylogenetic resolution (i.e., precision) [36]. Result obtained here via the Bayesian method (Figure A4) also show the drawbacks exemplified by very low values of posterior possibilities at key nodes (e.g., crown-group Priapulida (0.16), Nematoida + Panarthropoda (0.22), and many Cambrian worms such as stem-group Nematoida + Panarthropoda (below 0.1).

### 4.2. Eight Scalids around the First Circles: Morphological and Evolutionary Significance

The position of *Ercaivermis* close to the crown-group Priapulida seems to be largely influenced by the presence of eight scalids around the first circle [24,25,26], a character considered as symplesiomorphy for priapulids and well expressed in larval stages [26]. However, this feature is also present in other scalidophoran groups such as extant Loricifera and *Markuelia* (Cambrian–Ordovician). These scalids are not simple cuticular outgrowths but are innervated by eight nerve clusters, as seen in extant loriciferans [44]. A comparable relationship with the nervous system occurs in priapulids such as *Tubiluchus troglodytes* [45] although its eight scalids are innervated by 25 clusters [23]. In Kinorhyncha, ten scalids are present around the first introvert circle and are innervated by ten clusters of nerves originating from the circumoral brain [46,47].

These examples clearly show that the scalids present around the top circle of the introvert are more than such an external ornament but also have a sensory function. Their innervation, although variable in its wiring network, occurs in the three extant scalidophoran lineages (Priapulida, Loricifera, Kinorhyncha) [26,44,46,47] and is likely to have been present in their common ancestor. We hypothesize that the first scalid circle (whether the scalid number is 8 or 10) is a homologous sensory complex for the three groups and has therefore a potential evolutionary significance. “Eight scalids” can no longer be considered as a diagnostic feature of Priapulida since Loricifera also possesses this character. The presence of eight scalids in *Markuelia* (assumed stem-group Scalidophora [3,21]) may suggest that this character was inherited from an ancestral scalidophoran stock.

### 4.3. How Can We Recognize a Crown-Group Priapulida?

There is actually very little difference between early priapulids and the modern representatives of the group in terms of overall morphology (e.g., introvert, pharynx, cuticular ornament) and associated functional aspects [5,8,20]. All of them, including early Cambrian species, seem to have moved through their environment via muscle contractions and introvert eversion/inversion, and ingested food via a pharyngeal complex lined with teeth [48,49]. The basic morpho-functional features seem to have been remarkably stable over more than 500 million years. The apparent conservatism of the body plan of Priapulida might be linked to the environment occupied by these worms and their burrowing lifestyle [50,51], that both may have remained virtually unchanged since the Cambrian [52,53]. The apparent lack of post-Cambrian major innovations of the body plan within the group makes it difficult to define crown-group priapulids and to determine when it arose.

However, attempts have been made by authors to define a series of possible morphological diagnostic features such as: (1) introvert with external scalids (eight scalids around the first circle, 25 longitudinal rows of scalids); (2) well-developed pharynx with multidentate chitinous teeth (pentagonal symmetry); (3) neck between the introvert and annulated trunk; and (4) unpaired or paired caudal appendages (although not present in all taxa) [24,25,26]. Features (1) and (2) occur in the first and second loricate larvae in the first and second loricate larvae of *Priapulus caudatus*, respectively [54], and in meiofaunal priapulids such as *Tubiluchus* [24]. This set of characteristics raises a number of questions. As we have seen before, “eight scalids around the first circle” also occurs in loriciferans. Moreover, the neck is not unique to priapulids [17,23] and also occur in kinorhynchs [44] and loriciferans [32]. More importantly, three exceptions can be found in priapulids (see below) and make it difficult to identify a crown-group priapulid based on morphological features alone. For example, both larval and adult stages of *Meiopriapulus* [55,56], a meiofaunal species with a direct development [57], does have eight scalids in the first circle but a total of 150 longitudinal rows of scalids (from the fourth to twenty-second circles [55]). Moreover, the radial symmetry of its pharyngeal teeth is octoradial [25,56], thus contrasting with the overwhelmingly pentagonal symmetry of other extant priapulids. *Maccabeus*, a tubicolous priapulid from meiofaunal environments has 8, 25 and 16 scalids in the first, second and third circle, respectively and 25 in the remaining posterior circles [58]. The remarkably high number of scalids in the second and third circles may result from the merging of two adjacent circles [23,25].Despite these exceptions, the vast majority of crown-group Priapulida may share the following characters: (1) “25 longitudinal rows of scalids”; (2) “(8 + 9 + 8) scalid pattern for the first three circles”, and (3) “pentagonal arrangement of pharyngeal teeth”.

### 4.4. Symmetry in Fossil Priapulids

Although absent in some groups such as sponges and placozoans, radial and bilateral symmetry occurs in most present-day animal groups and is variously expressed both externally (e.g., overall shape, ornament) and internally (e.g., organs) [2,59]. Extant priapulids are by definition biradial animals, i.e., their adult body plan results from the combination of bilateral (e.g., nervous system, gonads, other aspects of their early development [24,25]) and radially symmetrical features (e.g., cuticular outgrowths such as scalids and pharyngeal teeth [23,59]). Developmental studies clearly indicate that these radial features superimpose a basically bilateral body plan [24,25,54]. Pentaradial symmetry largely prevails among modern priapulids and can be clearly seen in the arrangement of their pharyngeal teeth (fontal view showing pentagonal pattern) and the distribution of their external scalids that typically align in 20, 25, or 30 longitudinal rows [24,25,26]. Comparable symmetry patterns are known in numerous Cambrian stem-group or assumed crown-group priapulid worms such as *Yunnanpriapulus* [17], *Xiaoheiqingella* [17,18] and *Priapulites* [60], and stem-group priapulids such as *Selkirkia sinica* and *Sicyophorus* [21]. Different symmetry types co-existed in the Cambrian. For example, stem-group priapulids such *Eopriapulites* [29], *Shanscolex* [61], and *Mafangscolex* cf. *yunnanensis* [62] display hexaradial features (scalid rows). The discovery of octoradial symmetry (16 longitudinal rows of scalids) in *Ercaivermis* reinforces the hypothesis that symmetry patterns were more diverse in the early evolution of priapulids than they are today (see also [5,63]).

This is not an isolated case, and we also note that early cnidarians (e.g., Kuanchuanpu Formation; ca. 535 Ma; [64]) similarly developed more symmetry types (3, 4, 5) than can be found in nature today. For example, pentaradial symmetry is frequent among early Cambrian cnidarians whereas it is unknown in extant representatives of the group. This would suggest that the symmetry types that prevail today and characterize animal phyla may have resulted from natural selection through their evolution. McMenamin (2016) [65] hypothesized that morphogenetic evolution might be governed by nine laws, among them standardization or simplification. The decline of symmetry diversity that the evolutionary history of Cambrian priapulids suggests, i.e., the natural selection of one symmetry (e.g., pentaradial) type over others associated dynamically with the extinction of stem groups and rise of novel descendants, may indeed result from such a standardization process. However, uncertainties remain concerning the biological mechanisms and external drivers (e.g., extinction, see [63]) that would have controlled this hypothetical natural selection of symmetry patterns. Moreover, the impact of external symmetry patterns (e.g., distribution of scalid rows on introvert) on vital aspects of the function of these worms is particularly difficult to assess and would require experiments with extant species.

## 5. Conclusions

*Ercaivermis sparios* with 16 longitudinal rows of scalids (i.e., octoradial symmetry) differs markedly from most extant and extinct priapulid worms in which pentaradial symmetry prevails (e.g., 25 longitudinal scalid rows). Cladistic analyses resolve *Ercaivermis* as a stem-group priapulid. The first row of eight scalids that runs around the introvert of priapulids and loriciferans is linked to the nervous system and may be a character of particular importance in the phylogeny of scalidophorans, possibly present in their common ancestor. *Ercaivermis* do possess this character. Various symmetry types in external cuticular features (e.g., scalid rows) co-existed among Cambrian priapulid (penta-, hexa-, and octoradial) whereas a single type (pentaradial) prevails in the modern representatives of the group. This fossil evidence supports current evolutionary models in which standardization of symmetry patterns may occur as priapulid lineages evolve. The importance of symmetry of scalid rows must be interpreted with caution since its real impact on the animal’s vital functions remain to be clarified. Difficulties remain in determining the diagnostic characters of crown-group priapulids that may have appeared very early in the evolution of the group.

## Figures and Tables

**Figure 1 biology-12-01242-f001:**
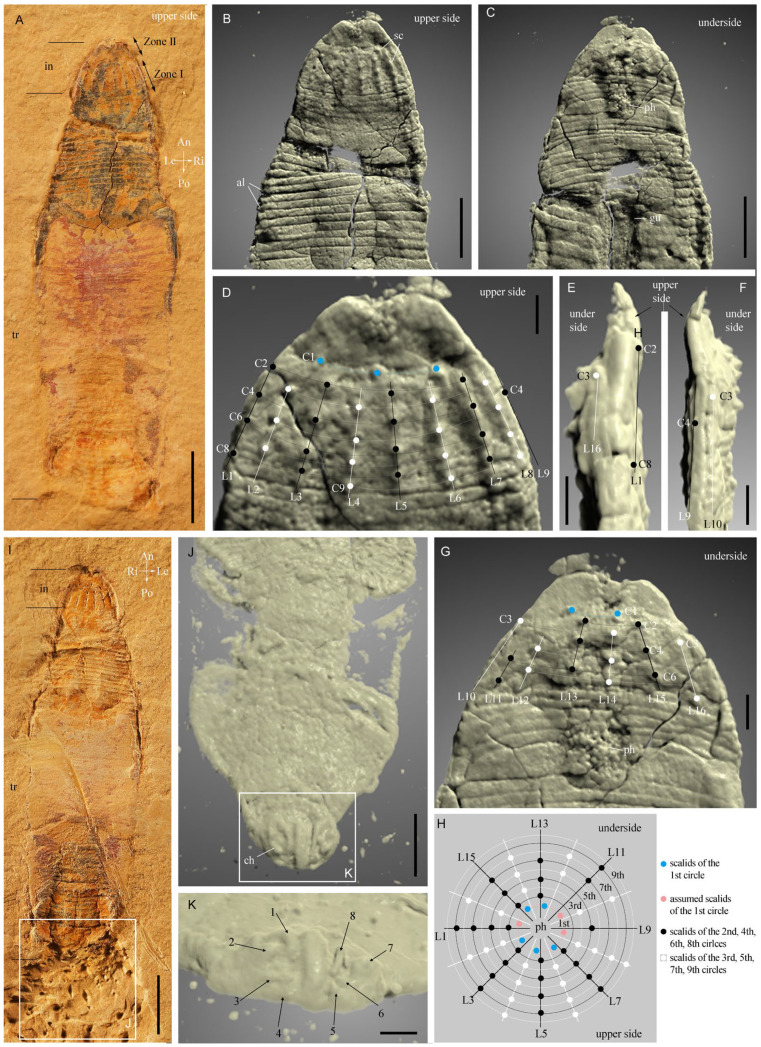
External morphology of *Ercaivermis sparios* gen. et sp. nov. from the early Cambrian of the Chengjiang Lagerstätte. (**A**) ELIEC-312A, part of holotype, general view showing body subdivision. (**B**–**G**) micro-CT images showing details of the introvert. (**B**–**D**) upper side showing the number of longitudinal rows and circles of scalids in Zone I, unarmed Zone II with imprints of pharyngeal teeth. (**E**,**F**) lateral views showing the edge of the compressed introvert. (**G**) underside of the specimen showing the incomplete distribution of scalids and the pharynx with teeth (Zone III). (**H**) diagram to show the octagonal distribution of scalids. (**I**) ELIEC-312B, counterpart of holotype. (**J**) circumanal region as revealed by micro-CT in (**I**). (**K**) close-up view of the four pairs of hooks around the anus; note their bilateral arrangement. Abbreviations: al, annulations; An, anterior; ch, caudal hook; gu, gut; in, introvert; Le, left; ph, pharynx; Po, posterior; Ri, right; sc, scalid; tr, trunk; C1~C9, 1st to 9th circle of scalids; L1~L16, 1st to 16th longitudinal row of scalids. Scale bars represent: 2 mm (**A**,**I**), 1 mm (**B**,**C**,**J**); 300 μm (**K**); 250 μm (**D**–**G**).

**Figure 2 biology-12-01242-f002:**
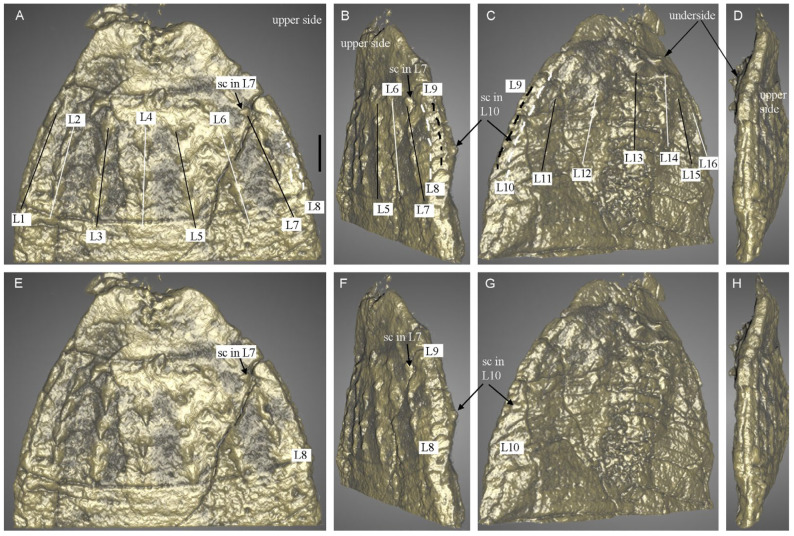
Scalid arrangement along the introvert of *Ercaivermis sparios* gen. et sp. nov. from the early Cambrian of the Chengjiang Lagerstätte, ELIEC-312A, part of the holotype. (**A**–**D**) front part (upper side), right-lateral view, back view (underside), and left-lateral view. Dashed lines in white and black indicate L8 in (**A**,**B**), L10 in (**C**) and L9 in (**B**,**C**), respectively. (**E**,**H**) same as (**A**–**D**) to show the scalids in L7 and L10 in (**E**–**G**) and upper side in (**H**). Same scale between (**E**–**H**) and (**A**–**D**). Scale bar represents 250 μm (**A**–**D**).

**Figure 3 biology-12-01242-f003:**
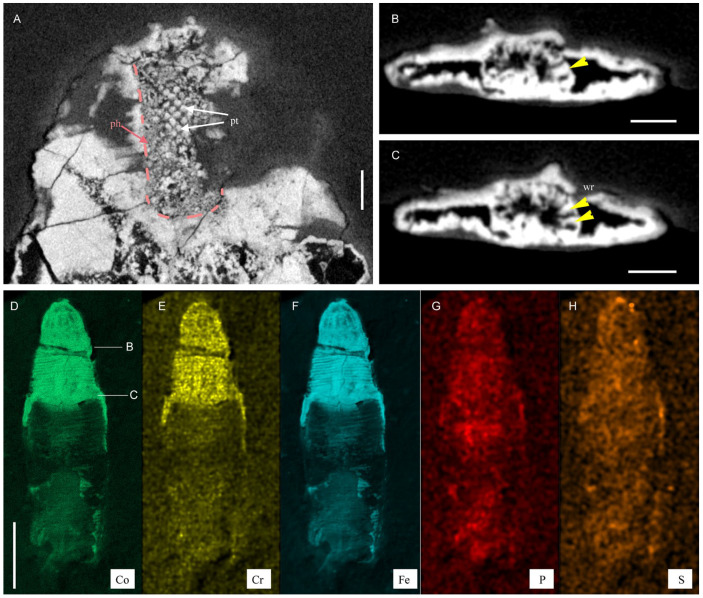
Internal anatomy and elemental mapping (XRF) of *Ercaivermis sparios* gen. et sp. nov. from the early Cambrian of the Chengjiang Lagerstätte, ELIEC-312A, part of the holotype. (**A**–**C**) micro-CT images. (**A**) longitudinal section showing the pharynx (pink dotted lines) and teeth. (**B**,**C**) transverse sections showing the gut with wrinkles (yellow arrowheads). See location of transverse virtual sections in (**D**). (**D**–**H**) elemental maps for Co, Cr, and Fe; note Fe-enrichment in cuticular remains; in contrast, P and S are enriched in the internal parts of the trunk. Abbreviations: ph, pharynx; pt, pharyngeal teeth; wr, wrinkle. Scale bars represent: 2 mm (**D**–**H**), 250 μm (**A**); 200 μm (**B**,**C**).

**Figure 4 biology-12-01242-f004:**
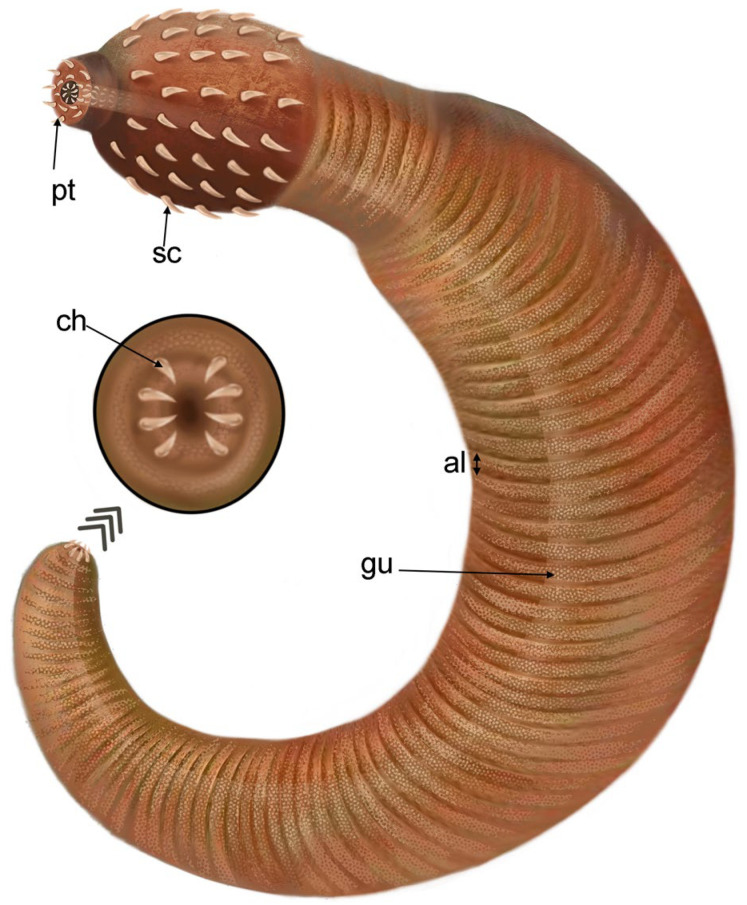
*Ercaivermis sparios* gen. et sp. nov. Artistic reconstruction showing body shape and circumanal area with hooks. Abbreviations: al, annulations; ch, caudal hook; gu, gut; pt, pharyngeal teeth; sc, scalid.

**Figure 5 biology-12-01242-f005:**
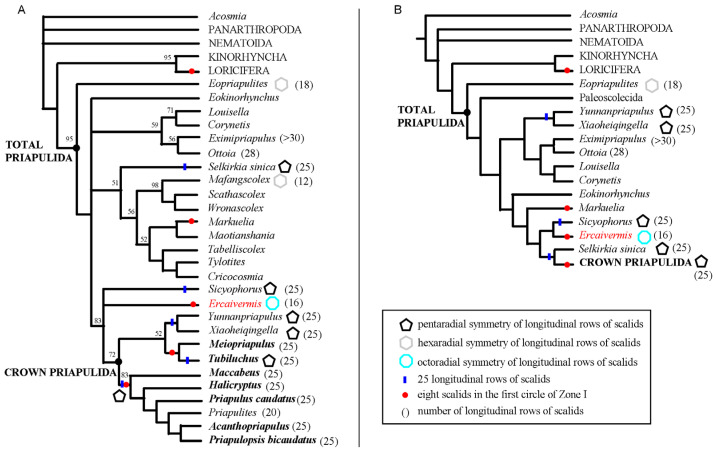
The phylogenetic position of *Ercaivermis*, scalid distribution and symmetry pattern indicated for some taxa. (**A**,**B**) simplified topologies recovered from maximum likelihood (IQ-TREE) (**A**) and implied weight (k ≥ 5) parsimony (TNT) (**B**) phylogenetic analyses. Numbers in (**A**) are bootstrap support values. See complete set of topologies in Figure A1, Figure A2, Figure A3 and Figure A4. Taxa in bold indicates the extant priapulid species.

## Data Availability

Specimens of *Ercaivermis* are housed in Northwest University, Xi’an (ELI), China. The computed tomography data of ELIEC-00312A and ELIEC-00312B are available in the Appendix A.

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
