# Peer review of "A New Chengjiang Worm Sheds Light on the Radiation and Disparity in Early Priapulida"

_biology, 2023, doi:10.3390/biology12091242_

Round 1
Reviewer 1 Report
Wang and colleagues describe a new genus and species of priapulan worm from the early Cambrian Chengjiang Lagerstätte, named Ercaivermis sparios, with important phylogenetic implications.
The Chengjiang Lagerstätte offers a unique window into early Cambrian ecosystems and each new species deepen and or often challenge our knowledge on certain groups, in this case, priapulans.
But, however worthy of publications the content and significance of this study are, the present manuscript, in my opinion, is not.
From the very first lines of the “Simple Summary” there are repetitions (e.g., “characterized by”), punctuation mistakes, typos, etc., some of them so evident that the impression is that the authors have not even re-read the text prior to the submission. Several parts of the manuscript are affected by the same problems, with some crass and easily avoidable mistakes. For instance, many taxon names are not in italics.
The language in itself necessitates extensive editing.
The “Simple Summary” is, by the way, not that “simple”, if it is meant for a general audience, as it includes geological, taxonomical, and anatomical terms without explanation in support (e.g., Cambrian Lagerstätte).
There also a lot of style and language inconsistencies (e.g., “Micro-CT” and “μCT”), or the apparently random use of inverted commas/quotation marks/apostrophes when describing and discussing characters and character states.
Collectively, all these factors hamper a clear understanding of several paragraphs and reiterate the feeling this is a hastily prepared manuscript, which is a pity, considering its potential significance.
At this stage, I have not dealt much with the properly scientific aspects of the manuscript, but there are some evident lacks, like the absence of any background on the locality (ok, Chengjiang is famous, but something should be said to contextualize this new species) or the fact that the “Supplementary Materials” are not accessible from the link provided by the authors, so that it is not possible to check any aspect of the analysis.
Extensive editing required.
Author Response
We thank you for your constructive critical comments and revised our MS accordingly. All modifications in the revised version are marked in red. Numbers (e.g. L-01) used in the following point-by-point reply refer to the corresponding line numbers in the revised MS.
Best regards,
All authors
From the very first lines of the “Simple Summary” there are repetitions (e.g., “characterized by”), punctuation mistakes, typos, etc., some of them so evident that the impression is that the authors have not even re-read the text prior to the submission. Several parts of the manuscript are affected by the same problems, with some crass and easily avoidable mistakes. For instance, many taxon names are not in italics.
Reply: We agree with you and correct these mistakes.
The language in itself necessitates extensive editing.
Reply: We agree with you and did our best to improve the language of the manuscript. Please note that none of us is a native speaker of English.
The “Simple Summary” is, by the way, not that “simple”, if it is meant for a general audience, as it includes geological, taxonomical, and anatomical terms without explanation in support (e.g., Cambrian Lagerstätte).
Reply: We agree with you that the “Simple Summary” should be simple and readily accessible to non-specialists. We rewrote it and removed all technical words (see revised version of the MS).
There also a lot of style and language inconsistencies (e.g., “Micro-CT” and “μCT”), or the apparently random use of inverted commas/quotation marks/apostrophes when describing and discussing characters and character states.
Reply: We agree with you and correct these mistakes.
At this stage, I have not dealt much with the properly scientific aspects of the manuscript, but there are some evident lacks, like the absence of any background on the locality (ok, Chengjiang is famous, but something should be said to contextualize this new species) or the fact that the “Supplementary Materials” are not accessible from the link provided by the authors, so that it is not possible to check any aspect of the analysis.
Reply: We agree with you and provided more details on the geological background (see Materials and preservation, lines 83-90) and replaced this new species into a broader palaeontological background. As for the “Supplementary Materials”, the link for accessing the supplementary materials has been reset (see line 375).
Reviewer 2 Report
Dear Authors
After reading his work I find it interesting and significant, especially for the use made of the inclusion of tomography in the visualization of fossil materials. I know that the images provided often lose sharpness when converted to pdf; however, I am concerned about the quality and readability of your Figure 1A. I suggest you present a larger version of your trees even if it takes several pages. In addition, in subsection E of this same figure, they must avoid that the blue box overlaps with the tree of subsection F, they must also try to frame in blue the entire moophyletic group. I think doing this with an image editor will be very simple.
Author Response
We thank you for your constructive critical comments and revised our MS accordingly. All modifications in the revised version are marked in red. Numbers (e.g. L-01) used in the following point-by-point reply refer to the corresponding line numbers in the revised MS.
Best regards,
All authors
After reading his work I find it interesting and significant, especially for the use made of the inclusion of tomography in the visualization of fossil materials. I know that the images provided often lose sharpness when converted to pdf; however, I am concerned about the quality and readability of your Figure 1A. I suggest you present a larger version of your trees even if it takes several pages.
Reply: We agree with you and re-formatted Figure A1 into four separate figures A1 to A4 (see Appendix lines 393-406).
In addition, in subsection E of this same figure, they must avoid that the blue box overlaps with the tree of subsection F, they must also try to frame in blue the entire monophyletic group. I think doing this with an image editor will be very simple.
Reply: We agree with you and correct these flaws, please see new appendix figures A1 to A4 (see Appendix lines 393-406).
Reviewer 3 Report
The perfectly preserved and thoroughtly described specimen of an Early Cambrian worm from the Chengjiang locality is of general interest and significantly increases knowledge of its taxonomic group. It definitely deserves publication.
However, the problematic aspect of the paper is that the concept of radial organisation of the body plan adapted by the authors is seriously distorted. The ‘body plan’ is the most general aspect of an animal anatomy that develops during its embryogenesis or larval development. In the case of Priapulus and related forms, their bodies are bilaterally symmetrical, which is clearly shown by the location of the central nerve cord. The larva, with its lorica, shows a biradial symmetry superimposed on the bilateral one. However, the disposition of scalids on the prosoma is apparently controlled by a low-rank developmental mechanisms, presumably of the same kind as those in, say, developing flowers or twig buds in plants. To claim that the number of scalids, different in particular circles, is an expression of radial symmetry does not make any sense. If accepted, such attitude would make hooks on the acanthocephalan (or the nematode Gnathostoma) proboscis evidence of their radial ground plans. Actually, in case of the new Chengjiang worm, its biradial symmetry component is clearly exposed at its anal end.
Such distortion of the basic biological concept must have its consequence in results of the cladistic analysis performed by the authors. Computer cladistics allows to assume that disposition of scalids is of value equal to any other anatomical trait. One may then receive expected topology of the cladogram by arranging matrix with an appropriate set of characters. A good example was offered recently by a paper published in the journal Nature, in which computer cladistic analysis allowed to interpret a Cambrian dasycladacean alga as the oldest known bryozoan. To claim that the number of scalids in particular circles on the priapulid prosoma is a character diagnosing phylum is equally strange (if not ridiculous). By analogy, the number of teeth in the radula should be used to diagnose the phylum Mollusca. Of course, I have no illusions that my comments will be seriously considered by the authors (and editors). It is their own responsibility to publish such an awkward evolutionary interpretations or not.
Author Response
We thank you for your constructive critical comments and revised our MS accordingly. All modifications in the revised version are marked in red. Numbers (e.g. L-01) used in the following point-by-point reply refer to the corresponding line numbers in the revised MS.
Best regards,
All authors
However, the problematic aspect of the paper is that the concept of radial organisation of the body plan adapted by the authors is seriously distorted. The ‘body plan’ is the most general aspect of an animal anatomy that develops during its embryogenesis or larval development. In the case of Priapulus and related forms, their bodies are bilaterally symmetrical, which is clearly shown by the location of the central nerve cord. The larva, with its lorica, shows a biradial symmetry superimposed on the bilateral one. However, the disposition of scalids on the prosoma is apparently controlled by a low-rank developmental mechanisms, presumably of the same kind as those in, say, developing flowers or twig buds in plants. To claim that the number of scalids, different in particular circles, is an expression of radial symmetry does not make any sense. If accepted, such attitude would make hooks on the acanthocephalan (or the nematode Gnathostoma) proboscis evidence of their radial ground plans. Actually, in case of the new Chengjiang worm, its biradial symmetry component is clearly exposed at its anal end.
Reply: We agree with the reviewer’s remarks that priapulids are biradial animals. i.e. they combine a bilateral organization and radially distributed features exemplified by scalids and pharyngeal teeth (typically 5-fold symmetry). Developmental studies show that radially symmetrical features superimpose the bilateral organization through the development from larval to adult stage. This basic information is essential to unspecialized readers and was missing in the first version of our MS. We added a few sentences to the revised version of the MS (see lines 323-341). The new worm described here has 16 rows of scalids around its trunk instead of 25 in the vast majority of extinct and extant priapulids. This unusual number of scalid rows gives this worm an 8-fold symmetrical appearance. Similarly, 8 scalids are found around the first circle of the introvert. We are aware that the expression of scalids may result from low-rank developmental mechanisms. However, we find it important to note that Ercaivermis differs from the majority of priapulid worms by its scalid distribution and symmetry expression. These differences do not result from intraspecific variations are likely to have evolutionary significance.
Such distortion of the basic biological concept must have its consequence in results of the cladistic analysis performed by the authors. Computer cladistics allows to assume that disposition of scalids is of value equal to any other anatomical trait. One may then receive expected topology of the cladogram by arranging matrix with an appropriate set of characters. A good example was offered recently by a paper published in the journal Nature, in which computer cladistic analysis allowed to interpret a Cambrian dasycladacean alga as the oldest known bryozoan. To claim that the number of scalids in particular circles on the priapulid prosoma is a character diagnosing phylum is equally strange (if not ridiculous). By analogy, the number of teeth in the radula should be used to diagnose the phylum Mollusca. Of course, I have no illusions that my comments will be seriously considered by the authors (and editors). It is their own responsibility to publish such an awkward evolutionary interpretations or not.
Reply: It would have been possible to assign a greater or lesser importance to a particular character (ex: number of scalids around introvert) in our cladistic analysis. Doing so is however a totally arbitrary choice that, to us, is not without risk. We are aware that cladistics is not a magic solution that can solve any phylogenetic problem. Results always need to be discussed. The points you raised are perfectly relevant (e.g. does it make sense to define a phylum on the basis of the number of scalids in a single circle?). We took your remarks into account and, accordingly, tried to enrich our discussion with a few sentences (see lines 244-254, 262-283). Please note that we are well aware of these problems (as are the editors of this scientific magazine). The number and morphology or teeth in mollusk radula vary between and within species. In contrast, the number and distribution pattern of scalids and pharyngeal teeth are very stable among extant priapulids (e.g. pentaradial distribution) and therefore potentially more reliable for phylogenetic analysis.
Reviewer 4 Report
The authors define the new Chengjiang taxon "Ercaivermis sparios". They argue that the lack of the standard 25 longitudinal scalid rows as seen in crown-group priapulans indicates that "Ercaivermis" is a stem priapulan, even though it has 8 scalids around the first introvert circle as typical for crown priapulans. From this analysis, the authors conclude that "symmetry patterns were more diverse among early priapulans than in extant representatives of the group." I generally agree with this conclusion, in fact, it is an example of the 3rd law of morphogenetic evolution, which involves standardization of patterning as lineages evolve (Dynamic Paleontology, 2016, Springer, p. 11). The revised manuscript should consider comparable evolutionary trajectories in kinorhynchs and related kinorhynch-like early Cambrian animals such as Eokinorhynchus rarus, and consider other cases where presumed rapid diversification is associated with more diverse symmetry patterns. Such an analysis will provide useful clues that can help us better understand the evolutionary dynamics of the Cambrian diversification event.
Comments on text:
34 fix improtant
67 add s to 'at odd'
238 italics for Priapulus caudatus
239 italics for Tubiluchus
315-314 The coexistence of penta-, hexa- and octoradial symmetries in Cambrian stem priapulians provide criterion examples of the 2nd and 3rd laws of morphogenetic evolution (see reference above)
none
Author Response
We thank you for your constructive critical comments and revised our MS accordingly. All modifications in the revised version are marked in red. Numbers (e.g. L-01) used in the following point-by-point reply refer to the corresponding line numbers in the revised MS.
Best regards,
All authors
The authors define the new Chengjiang taxon "Ercaivermis sparios". They argue that the lack of the standard 25 longitudinal scalid rows as seen in crown-group priapulans indicates that "Ercaivermis" is a stem priapulan, even though it has 8 scalids around the first introvert circle as typical for crown priapulans. From this analysis, the authors conclude that "symmetry patterns were more diverse among early priapulans than in extant representatives of the group." I generally agree with this conclusion, in fact, it is an example of the 3rd law of morphogenetic evolution, which involves standardization of patterning as lineages evolve (Dynamic Paleontology, 2016, Springer, p. 11). The revised manuscript should consider comparable evolutionary trajectories in kinorhynchs and related kinorhynch-like early Cambrian animals such as Eokinorhynchus rarus, and consider other cases where presumed rapid diversification is associated with more diverse symmetry patterns. Such an analysis will provide useful clues that can help us better understand the evolutionary dynamics of the Cambrian diversification event.
Reply: The standardization of patterning as lineages evolve is a very interesting hypothesis that require testing from fossil evidence. We agree with you that we should stress on this important evolutionary mechanism. Cambrian priapulids may indeed indicate that symmetry patterns were more diverse in the early stages of the group and underwent standardization later on in the course of evolution. We added a few sentences and briefly discussed this topic with reference to Dynamic Paleontology, 2016 (see lines 347-351). We agree that discussing the evolutionary trajectories of other scalidophoran groups (e.g. Kinorhyncha, Loricifera) is essential and may lead to the identification of general trends. However, this would require detailed analyses of fossil material that we prefer to reserve for another article.
Comments on text:
34 fix improtant
Reply: We revised the sentence and removed this typo.
67 add s to 'at odd'
Reply: OK, done. Similar mistakes have been revised in the text.
238 italics for Priapulus caudatus
Reply: OK, done. We checked and corrected similar problems throughout the MS.
239 italics for Tubiluchus
Reply: OK, done. Similar mistakes have been revised in the text.
315-314 The coexistence of penta-, hexa- and octoradial symmetries in Cambrian stem priapulians provide criterion examples of the 2nd and 3rd laws of morphogenetic evolution (see reference above)
Reply: We agree with you and referred to these important laws of morphogenetic evolution (see lines 347-351).
Reviewer 5 Report
Wang et al. describe a Cambrian priapulan from Chengjiang, based on a single specimen whose extensive pyritization allows three-dimensional reconstruction through microCT analysis. This reveals details, particularly of the posterior end, that are not available from inspection of the plane of splitting. The authors also provide a detailed reconstruction of the dentition of the introvert.
The wider implications hang on this introvert reconstruction, and particularly on the identification of sixteen longitudinal rows of teeth (in contrast to the 20/25 rows of extant priapulans). Although certain aspects of the introvert are well documented by the figures, however, the incomplete preservation, alongside inconsistencies in the presentation, do not unambiguously support the authors’ inference of exactly 16 introvert rows.
Nonetheless, the new taxon does not obviously belong to the crown group, and is likely correctly interpreted as a stem-group taxon – no great surprise, as the earliest bona fide members of the crown group do not arise until the Carboniferous, or perhaps later. The authors contend that this allows them to discuss morphological characteristics that might define the crown group, but these arguments are unsound: for instance, the proposal that “eight scalids in the first introvert circle” is diagnostic of crown-Priapulida is demonstrably false, as this characterizes the new specimen (which does not belong to the crown group) as well as Loriciferans. The authors also make some broad claims about the diversity/disparity of Cambrian priapulans, but do not provide any quantitative analysis to support these claims.
Taken together, then, this is an interesting fossil that represents a new species and potentially fits into the story of priapulan evolution; the fossil itself merits description and publication. However, the attempts to draw broader significance from the fossil are not well founded and rely on too much speculation. Either these should be fully substantiated and shown to be robust, with speculation treated with appropriate caution; or the paper should be pared back to be primarily descriptive in scope.
DESCRIPTION
The fundamental question in the description of this species pertains to the arrangement of the introvert dentition. The authors’ interpretation of the fossil material is reasonably clear from the figures, but the markings somewhat obscure the fossil anatomy itself (which is perhaps available in the supplementary information?). If space permits, it would be helpful to display the unannotated specimen alongside an annotated panel. Better still would be to segment the model itself, which would allow the authors to render each longitudinal row in a distinct colour that is consistent between figures. Doing this with the 3D data rather than by annotation of 2D renders would increase the confidence that each row is correctly identified, and that the same landmarks are being used consistently. At present there are a number of errors in the interpretation that place significant doubt that the number of scalid rows has been counted correctly. For example:
-
The large scalid at the top of L8 in Fig. 2A is identified as belonging to L7 in Fig. 2B, suggesting that this row has been counted twice and the introvert in fact has 15 rows (and thus pentaradial symmetry)
-
Fig. 1H and Fig. 2 display no scalids in L10. In the absence of scalids, there seems to be no evidence for the existence of this row (which would point to 15, rather than 16, rows of scalids).
-
In contradiction, Fig. 1G denotes a scalid in position C3 on line L10. (This scalid is not visible in the presented figure, possibly because it is concealed by the white spot marking its location.)
These inconsistencies raise some concerns about the validity and accuracy of the authors’ depiction of scalid layout (Fig. 1H). But even if this figure does correctly depict the arrangement of preserved scalids, the case remains that preservation is incomplete. As such, it is hard to see how the interpretation of 8+8+8 sclerites in the first three rows can be justified over the more parsimonious 8+9+8 typical of extant priapulans when the authors only observe 5+6+6 scalids. This interpretation takes a central role in the remainder of the manuscript, but ultimately cannot be demonstrated with confidence given the non-preservation of a quarter of the available scalids.
A number of secondary details of the description and interpretation would also benefit from additional clarification.
Circlet 1 is out of alignment with regards to the subsequent introvert rows. Please justify its inclusion within Zone I, over an identification with the circumoral spines that ring the base of Zone II in most Cambrian priapulomorphs.
The “cluster of tiny pharyngeal teeth” in Zone II require annotation on the figures. I think that the authors intend to communicate that these are the pharyngeal teeth of Zone III, which are displayed through the thin/eroded cuticle of Zone II – which indeed strikes me as the most plausible interpretation, given the spacing of the teeth and their restriction to the medial axis. However, the text makes it sound like the teeth belong to Zone II (which is unarmed in this specimen, as in other Cambrian priapulomorphs).
Based on the scale bar in Fig. J, the trunk appears to be around 3 mm wide along its length, except for the terminal region which is <1 mm in width; I don’t see a basis for the 1.5 mm figure quoted in the text.
Please indicate how the “wrinkles” interpreted in the gut can be distinguished from pharyngeal teeth, and also denote the position of the teeth on the pharynx cross-sections. How can wrinkles be distinguished from e.g. diverticulae? The text should make clear the interpreted orientation of the wrinkles, which are not illustrated in Fig. 4 as promised in the text.
DISCUSSION
The underlying motivation of the discussion seems to be the identification of characters that define the priapulan crown group to the exclusion of stem group representatives. It is difficult to see what utility such a set of characters might display; and nor might we expect such a set of character to exist, given that the crown node is unlikely to be associated with any morphological innovation: the crown group comes about by extinction and is defined by the relationships of extant taxa; and the morphology that characterizes the crown will accumulate incrementally along the stem lineage.
The lack of clear taxonomic patterns in dentition within extant priapulans surveyed in the discussion somewhat suggests that these features are not particularly useful as phylogenetic characters, and hence that it is unwise to attach too much phylogenetic significance to them. It is not clear how fossils can contribute to the identification of commonalities between extant taxa (especially where there is uncertainty in the actual number of scalids in particular fossil taxa). But there is a fundamental contradiction in proposing that “ ‘Eight scalids around the first introvert circle’ seems to be an improtant [sic] and typical diagnostic feature of the crown-group Priapulida" [...] “found exclusively in the crown-group Priapulida” and the recognition of this character state in a taxon attributed to stem Priapulida.
The identification of the priapulan crown group has been discussed at some length by Budd & Jensen (2000) here, and even if dated, it is surprising not to see this pertinent discussion used as a starting point for the discussion.
Insofar as the authors consider the dynamics and diversity of the priapulan stem, they might also consider referring to work on the general dynamics of stem and crown groups (e.g. Budd & Mann 2020, Science Advances) as well as specific studies on the disparity of priapulans (e.g. Wills et al. 2012); does the new data really move the discussion on, or does it exemplify trends already known? It would be reasonably straightforward to incorporate the new taxon in the Wills et al. dataset to demonstrate whether it really does contribute to our picture of priapulomorph disparity in the Cambrian.
A couple of specific points in the discussion warrant revision:
“The evolutionary mechanisms and possible drivers that led to this reduction or canalization remain unknown.” – false; this was a function of non-pentaradial forms going extinct. It is practically inevitable that stem groups exhibit some morphologies that are not found in the crown group, as the crown group inherits its characters from a single ancestor whose contemporaries all belonged to the stem group.
The authors should also take care to distinguish the symmetry of the introvert from the symmetry of the organism as a whole. Adrianov & Malakhov (2001, J Morph) emphasize that priapulans exhibit “a combination of several radial symmetries: pentaradial symmetry of the teeth, octaradial symmetry of the primary scalids, 25-radial symmetry of scalids, biradial symmetry of the neck, and biradial and decaradial symmetry of the trunk”. The description (e.g. in Fig. 5) of crown priapulans as exhibiting only pentaradial symmetry is an over-simplification to the point of being misleading.
PRESENTATION
A word processing error seems to have garbled the opening sentence of the Simple Summary.
The four pairs of posterior hooks are somewhat obscured by the overprinted text in Fig. 1k; perhaps the numbers could be printed further to one side and lined by a narrow connecting line, so the outline of the hooks themselves can be observed? It is also unclear whether additional pairs of hooks may occur on the obverse of the specimen, as angled in Fig. 1k. What is the medial ridge, and why is this not included in the reconstruction in Fig. 4?
Figures: what is the significance of the background gradient? A solid black [/white] background would be easier to interpret.
Elemental maps: I am not clear what purpose these serve; the distributions of all elements figured seem simply to match that of Fe, presumably as they are all associated with the original pyrite. It might be more instructive to display a single map of iron distribution (if even this is necessary – as the distribution of pyrite corresponds exactly to the dark regions already visible in the reflected-light photographs) alongside bar charts comparing the relevant concentrations of the elements under examination in the pyritized region and in the matrix.
Fig. 4. The two rings of 8 scalids drawn surrounding the mouth opening are not discussed or described in the text; do these belong to Zone II?
Fig. 5. Please check that the colour scheme in the key matches that used in the figure.
Table:
- Is it really the case that the scalids can be measured to this level of precision, given how poorly defined their margins are in the CT scan? This data seems spurious and I do not see that it serves any practical or taxonomic value, particularly as the specimen is displayed with a scale bar in Fig. 1 and the raw data are apparently available in the supplementary information. (It was not possible to view the SI as this has not been published on FigShare and no working link for reviewers was provided.)
- How can Nr take a non-integer value?
Pixelation and a tiny font size makes Figure A1 impossible to read; the authors should consider a more instructive way of presenting these results and making them digestible to readers.
Please italicize the names of genera in the text and in references.
PHYLOGENETIC ANALYSIS
The data underlying the phylogenetic analysis are not available to the reviewers so cannot be evaluated. These must be properly scrutinized by reviewers before the work can be considered for publication; the authors should provide a working link to the FigShare repository, or provide the SI in some other format.
It seems likely that the coding of the new taxon will need to be updated to reflect the ambiguity in interpretation discussed above – and that the phylogenetic position may change accordingly.
Even then there are limits to how far these results should be relied upon. Correction for inapplicable data has been shown to substantially impact the results of analysis of the underpinning dataset in ref. 28 (see Smith & Dhungana 2022, JGS) so it would be instructive to compare the Fitch parsimony results with analysis performed after correcting for inapplicable data, either using the implementation recently added to TNT or the approximation implemented in Morphy/TreeSearch.
The analytical parameters are also questionable; k = 1 implements clique analysis and should never be used in a parsimony setting; k <= 3 has been shown to typically produce less accurate trees. K = 10 is a widely recommended value and would be much more instructive. Please justify the selection of concavity constants used with reference to appropriate literature.
The text implies that all phylogenetic analyses place the new taxon in stem-Priapulida, but the equal weights results appear to show it in the crown group, with a crown group position also potentially included in the set of most parsimonious trees with k = 3 (depending on what topologies underpin the relevant polytomy).
Does the phylogenetic analysis not provide a basis for testing the authors’ statement that “the occurrence of '8 scalids around the first circle' seems not homologous between Priapulida and Loricifera.”? The text in any case contradicts this statement; line 295 reports “found exclusively in the crown-group Priapulida”. It is also necessary to discuss which Cambrian taxa this character can be scored unequivocally for, as it requires exceptional preservation to be able to distinguish and count this first circlet precisely. Is the state really known with confidence outside extant taxa?
Generally good but some errors in grammar and word choice (e.g. "uneasy")
Author Response
We thank you for your constructive critical comments and revised our MS accordingly. All modifications in the revised version are marked in red. Numbers (e.g. L-01) used in the following point-by-point reply refer to the corresponding line numbers in the revised MS.
Best regards,
All authors
The wider implications hang on this introvert reconstruction, and particularly on the identification of sixteen longitudinal rows of teeth (in contrast to the 20/25 rows of extant priapulans). Although certain aspects of the introvert are well documented by the figures, however, the incomplete preservation, alongside inconsistencies in the presentation, do not unambiguously support the authors’ inference of exactly 16 introvert rows.
Reply: The longitudinal rows of scalids are preserved well on the upper side but sparse on the underside. Even though obscure longitudinal rows of scalids can be discernible. Total 16 longitudinal rows of scalids are numbered (Figure 1). To make sure the accuracy of the number of the longitudinal rows of scalid, we measure the parameters to calculate it. The calculated value is 15.6, meaning that the longitudinal rows of scalids is either 15 or 16. Because of the specimen compression, it results that the measured value of Wp smaller than the true one. Based on the formula above, the true value of Wp (i.e. larger than measured value of Wp) will lead to the value of x close to 16 (i.e. Nri).(see lines 234-238).
Nonetheless, the new taxon does not obviously belong to the crown group, and is likely correctly interpreted as a stem-group taxon – no great surprise, as the earliest bona fide members of the crown group do not arise until the Carboniferous, or perhaps later. The authors contend that this allows them to discuss morphological characteristics that might define the crown group, but these arguments are unsound: for instance, the proposal that “eight scalids in the first introvert circle” is diagnostic of crown-Priapulida is demonstrably false, as this characterizes the new specimen (which does not belong to the crown group) as well as Loriciferans. The authors also make some broad claims about the diversity/disparity of Cambrian priapulans, but do not provide any quantitative analysis to support these claims.
Reply: We agree with you that “8 scalids in the first introvert circlet” is a character found in extant priapulids and loriciferan worms as well and cannot be used as a diagnostic feature of Priapulida. We made it clear in the revised version of our MS (see lines 261-265). However, it is uncertain whether this character is homologue in both groups.
The morphological diversity/disparity of Cambrian priapulids has been studied by Wills et al. (1998, 2012) via quantitative analyses. Our approach is different and equally valuable and based on personal observations (not data from literature). We are using an exceptionally preserved fossil material that provides detailed information on a new worm that reveals an unusual scalid pattern and symmetry mode. Our results simply question the fact that symmetry patterns may have been more diverse in the early evolutionary history of Priapulida. We don’t think that it is “a broad claim” as you wrote. Instead, it is, to our eyes, an interesting hypothesis that is worth being mentioned in our discussion. More fossil data will hopefully tell us whether Ercaivermis is an exception or represents an alternative symmetry mode among Cambrian priapulids.
Taken together, then, this is an interesting fossil that represents a new species and potentially fits into the story of priapulan evolution; the fossil itself merits description and publication. However, the attempts to draw broader significance from the fossil are not well founded and rely on too much speculation. Either these should be fully substantiated and shown to be robust, with speculation treated with appropriate caution; or the paper should be pared back to be primarily descriptive in scope.
Reply: Although we toned down some speculative statements, we think that our hypothesis of priapulid disparity (incl. symmetry modes) being higher in the early Cambrian than nowadays is worth being discussed (see above) and should not be removed (see lines 319-353).
DESCRIPTION
The fundamental question in the description of this species pertains to the arrangement of the introvert dentition. The authors’ interpretation of the fossil material is reasonably clear from the figures, but the markings somewhat obscure the fossil anatomy itself (which is perhaps available in the supplementary information?).
Reply: We agree with you and made necessary improvements (see Figures 1, 2).
If space permits, it would be helpful to display the unannotated specimen alongside an annotated panel.
Reply: We agree with you. The unannotated images of Fig. 1D, G correspond to Fig. 1B, C, respectively. In Figure 2, we add unannotated images (E to H) corresponding to those of A to D.
Better still would be to segment the model itself, which would allow the authors to render each longitudinal row in a distinct colour that is consistent between figures. Doing this with the 3D data rather than by annotation of 2D renders would increase the confidence that each row is correctly identified, and that the same landmarks are being used consistently.
Reply: It is an excellent suggestion. Thank you. However, it would result in extensive changes in our figures; we prefer not to do it this time but keep your idea for further studies.
At present there are a number of errors in the interpretation that place significant doubt that the number of scalid rows has been counted correctly. For example: The large scalid at the top of L8 in Fig. 2A is identified as belonging to L7 in Fig. 2B, suggesting that this row has been counted twice and the introvert in fact has 15 rows (and thus pentaradial symmetry)
Reply: We check it carefully. The so-called “large scalid” you said lies definitively in L7 (see images below). L8 is separated and deviates from the “large scalid”.
Fig. 1H and Fig. 2 display no scalids in L10. In the absence of scalids, there seems to be no evidence for the existence of this row (which would point to 15, rather than 16, rows of scalids).
Reply: Please see the images below to show the L10 with a scalid C3 (white dot) that is a little of obscure, but it exists.
In contradiction, Fig. 1G denotes a scalid in position C3 on line L10. (This scalid is not visible in the presented figure, possibly because it is concealed by the white spot marking its location.)
Reply: See explanations above.
These inconsistencies raise some concerns about the validity and accuracy of the authors’ depiction of scalid layout (Fig. 1H). But even if this figure does correctly depict the arrangement of preserved scalids, the case remains that preservation is incomplete. As such, it is hard to see how the interpretation of 8+8+8 sclerites in the first three rows can be justified over the more parsimonious 8+9+8 typical of extant priapulans when the authors only observe 5+6+6 scalids. This interpretation takes a central role in the remainder of the manuscript, but ultimately cannot be demonstrated with confidence given the non-preservation of a quarter of the available scalids.
Reply: We admit that our specimen is incompletely preserved. The figure below is a diagrammatic representation of its scalid arrangement. There are 16 longitudinal rows of scalids arranged in quincunx. Eight scalids are likely to occur on the 2nd to 9th circle which is consistent with 16 longitudinal rows of scalids.
Five scalids (blue dots in diagram below) can be seen in the first circle. The position and number of the three remaining scalids (pink dots) are inferred. We assume that each of the five scalids around the first circle represents a point in the radii (red arrows, below figure). There are five radii, i.e. radiating lines drawn from the scalids of the first row. Every two longitudinal rows of scalids are between every two radii. According to this configuration, three additional scalids are inferred (pink dots in Fig. 1H, see also figure below). Thus, a total of 8 scalids is likely to have occurred around the first circle. Black and white dots represent the scalids in 3rd, 5th, 7th, 9th and 2nd, 4th, 6th, 8th circles, respectively.
A number of secondary details of the description and interpretation would also benefit from additional clarification.
Reply: OK, some modifications (see main text in red) have been done according to your comments.
Circlet 1 is out of alignment with regards to the subsequent introvert rows. Please justify its inclusion within Zone I, over an identification with the circumoral spines that ring the base of Zone II in most Cambrian priapulomorphs.
Reply: This is indeed an important remark and has been clarified clearly (see lines 180-182). Interestingly, comparable offset of the first circle also occurs in extant priapulids (see diagrams made by Adrianov and Malakhov 2001). The reason for this offset remains unknown.
The “cluster of tiny pharyngeal teeth” in Zone II require annotation on the figures. I think that the authors intend to communicate that these are the pharyngeal teeth of Zone III, which are displayed through the thin/eroded cuticle of Zone II – which indeed strikes me as the most plausible interpretation, given the spacing of the teeth and their restriction to the medial axis. However, the text makes it sound like the teeth belong to Zone II (which is unarmed in this specimen, as in other Cambrian priapulomorphs).
Reply: Yes, we are also aware of this problem and made the necessary corrections (see 186-190).
Based on the scale bar in Fig. J, the trunk appears to be around 3 mm wide along its length, except for the terminal region which is <1 mm in width; I don’t see a basis for the 1.5 mm figure quoted in the text.
Reply: We measured the terminal region again, its width is ca. 1.2 mm.
Please indicate how the “wrinkles” interpreted in the gut can be distinguished from pharyngeal teeth, and also denote the position of the teeth on the pharynx cross-sections. How can wrinkles be distinguished from e.g. diverticulae? The text should make clear the interpreted orientation of the wrinkles, which are not illustrated in Fig. 4 as promised in the text.
Reply: In extant priapulids (e.g. Priapulus), the pharynx is made of strong muscles and is never wrinkled. In contrast, the gut that follows the pharynx posteriorly has a very different structure and is made of non-cuticular endodermal epithelium. By definition, this gut tends to increase the surface of exchange in relation to its assimilating function. The gut of preserved specimens of Priapulids has a strongly wrinkled appearance that probably result from originally “wrinkled” cell layers and artefacts (tissue retraction) due to chemical preservation (typically, ethanol tends to retract tissues).
DISCUSSION
The underlying motivation of the discussion seems to be the identification of characters that define the priapulan crown group to the exclusion of stem group representatives. It is difficult to see what utility such a set of characters might display; and nor might we expect such a set of character to exist, given that the crown node is unlikely to be associated with any morphological innovation: the crown group comes about by extinction and is defined by the relationships of extant taxa; and the morphology that characterizes the crown will accumulate incrementally along the stem lineage.
Reply: Indeed, it is extremely difficult to determine which characters define the crown-group priapulids. There seems to be no important innovation in the body plan that would make a major difference between stem and crown group priapulids (see changes in lines 282-316).
The lack of clear taxonomic patterns in dentition within extant priapulans surveyed in the discussion somewhat suggests that these features are not particularly useful as phylogenetic characters, and hence that it is unwise to attach too much phylogenetic significance to them.
Reply: We agree with you the significance of scalid number and distribution is questionable (see lines 271-280). However, these characters have long been used in phylogenetic analyses (e.g. 25 five longitudinal rows of scalids; see Wills et al. 1998, 2012, Harvey et al. 2010, Shi et al. 2021, Howard et al. 2021). Most of these characters occur in many species and do not result from simple morphological variations with no evolutionary significance. Similarly, “8 scalids around the first circle” is not a diagnostic feature of Priapulida since this character also occurs in Loricifera. However, it cannot be considered as an anecdotal character with no evolutionary importance.
It is not clear how fossils can contribute to the identification of commonalities between extant taxa (especially where there is uncertainty in the actual number of scalids in particular fossil taxa).
Reply: Yes, there is some degree of uncertainty concerning the number of scalids and we don’t try to hide it (see lines 172-184). We know no fossil material where part of the anatomical information is not missing. Again, we think that our fossil specimen displays atypical morphological characters that are worth being shown and interpreted.
But there is a fundamental contradiction in proposing that “ ‘Eight scalids around the first introvert circle’ seems to be an improtant [sic] and typical diagnostic feature of the crown-group Priapulida" [...] “found exclusively in the crown-group Priapulida” and the recognition of this character state in a taxon attributed to stem Priapulida.
Reply: We agree with you and reconsidered the significance of this character which occurs in at least two scalidophoran groups (Priapulida and Loricifera). It cannot be considered as a diagnostic feature of crown-group Priapulida. We rewrote a large part of our discussion on crown-group Priapulida (see lines 260-280).
The identification of the priapulan crown group has been discussed at some length by Budd & Jensen (2000) here, and even if dated, it is surprising not to see this pertinent discussion used as a starting point for the discussion.
Reply: we agree with you and integrated Budd & Jensen’s discussions in our revised version (see lines 49, 283-285)
Insofar as the authors consider the dynamics and diversity of the priapulan stem, they might also consider referring to work on the general dynamics of stem and crown groups (e.g. Budd & Mann 2020, Science Advances) as well as specific studies on the disparity of priapulans (e.g. Wills et al. 2012); does the new data really move the discussion on, or does it exemplify trends already known? It would be reasonably straightforward to incorporate the new taxon in the Wills et al. dataset to demonstrate whether it really does contribute to our picture of priapulomorph disparity in the Cambrian.
Reply: we agree with you that our results have to discussed in the light of evolutionary scenarios proposed by other authors (e.g. Budd & Mann 2020, Science Advances and Wills et al. 2012). Some text is added to our discussions (see 49, 283-285, 355-338).
A couple of specific points in the discussion warrant revision:
“The evolutionary mechanisms and possible drivers that led to this reduction or canalization remain unknown.” – false; this was a function of non-pentaradial forms going extinct. It is practically inevitable that stem groups exhibit some morphologies that are not found in the crown group, as the crown group inherits its characters from a single ancestor whose contemporaries all belonged to the stem group.
Reply: We agree with you and removed this confusing sentence.
The authors should also take care to distinguish the symmetry of the introvert from the symmetry of the organism as a whole. Adrianov & Malakhov (2001, J Morph) emphasize that priapulans exhibit “a combination of several radial symmetries: pentaradial symmetry of the teeth, octaradial symmetry of the primary scalids, 25-radial symmetry of scalids, biradial symmetry of the neck, and biradial and decaradial symmetry of the trunk”. The description (e.g. in Fig. 5) of crown priapulans as exhibiting only pentaradial symmetry is an over-simplification to the point of being misleading.
Reply: We agree with you. In Fig. 5, we only focus on the symmetry of scalids on the introvert and correct the interpretation of the legends. It is true that all stem and crown species of priapulids are bilateral symmetry in gross morphology.
PRESENTATION
A word processing error seems to have garbled the opening sentence of the Simple Summary.
Reply: OK, such things have been corrected, see revised version.
The four pairs of posterior hooks are somewhat obscured by the overprinted text in Fig. 1k; perhaps the numbers could be printed further to one side and lined by a narrow connecting line, so the outline of the hooks themselves can be observed? It is also unclear whether additional pairs of hooks may occur on the obverse of the specimen, as angled in Fig. 1k. What is the medial ridge, and why is this not included in the reconstruction in Fig. 4?
Reply: OK, such things have been corrected, see revised version.
Figures: what is the significance of the background gradient? A solid black [/white] background would be easier to interpret.
Reply: No especially significance. It is background when save the images from the software DragonFly.
Elemental maps: I am not clear what purpose these serve; the distributions of all elements figured seem simply to match that of Fe, presumably as they are all associated with the original pyrite. It might be more instructive to display a single map of iron distribution (if even this is necessary – as the distribution of pyrite corresponds exactly to the dark regions already visible in the reflected-light photographs) alongside bar charts comparing the relevant concentrations of the elements under examination in the pyritized region and in the matrix.
Reply: The preservation of the specimen is discussed in “Material and methods”. Fe-enrichment is due to the presence of Fe-oxides (originally pyrite) as in other Chengjiang fossils. Although Co and Cr maps reveal no important information (these elements are commonly present in pyrite), P and S maps are worth being shown. The presence of S indicates that pyrite is still present. P may be associated to organic remains.
Fig. 4. The two rings of 8 scalids drawn surrounding the mouth opening are not discussed or described in the text; do these belong to Zone II?
Reply: They are assumed pharyngeal teeth expressed on the partially everted pharynx in Fig. 4, see revised figure.
Fig. 5. Please check that the colour scheme in the key matches that used in the figure.
Reply: OK, done.
Table:
- Is it really the case that the scalids can be measured to this level of precision, given how poorly defined their margins are in the CT scan?
Reply: Unfortunately, the resolution of our Micro-CT is relatively poor. However, measurements can be made, for example between the center of two adjacent scalids. We agree that the conditions are not optimal. However, our measurements indicate that 16 longitudinal rows of scalids occur along the introvert.
This data seems spurious and I do not see that it serves any practical or taxonomic value, particularly as the specimen is displayed with a scale bar in Fig. 1 and the raw data are apparently available in the supplementary information. (It was not possible to view the SI as this has not been published on FigShare and no working link for reviewers was provided.)
Reply: it was our mistake to provide an invalid web link to access the “Supplementary Material”. A new link is now available (see line 372).
- How can Nr take a non-integer value?
Reply: The data were calculated by a formula that will result in a non-integer value. That is plausible. A non-integer value of 15.6 of Nrm (revised abbreviation). An integer value is given (Nri, see revised version, lines 228-237).
Pixelation and a tiny font size makes Figure A1 impossible to read; the authors should consider a more instructive way of presenting these results and making them digestible to readers.
Reply: OK, done.
Please italicize the names of genera in the text and in references.
Reply: OK, done.
PHYLOGENETIC ANALYSIS
The data underlying the phylogenetic analysis are not available to the reviewers so cannot be evaluated. These must be properly scrutinized by reviewers before the work can be considered for publication; the authors should provide a working link to the FigShare repository, or provide the SI in some other format.
Reply: it was our mistake to provide an invalid web link to access the “Supplementary Material”. A new link is now available (see line 372)
It seems likely that the coding of the new taxon will need to be updated to reflect the ambiguity in interpretation discussed above – and that the phylogenetic position may change accordingly.
Reply: Thank you for this interesting suggestion.
Even then there are limits to how far these results should be relied upon. Correction for inapplicable data has been shown to substantially impact the results of analysis of the underpinning dataset in ref. 28 (see Smith & Dhungana 2022, JGS) so it would be instructive to compare the Fitch parsimony results with analysis performed after correcting for inapplicable data, either using the implementation recently added to TNT or the approximation implemented in Morphy/TreeSearch.
Reply: We agree with your comments on the inapplicable data. However, when dealing with the inapplicable data of Shi et al. (2021) by Smith and Dhungana (2022) under the BGS, the result showed a polychotomy of most priapulan-like worms. One can image that when analyzing the dataset inherited from Shi et al. (2021) with Ercaivermis, it would produce the similar result as that shown in Smith and Dhungana (2022). However, such result would contribute nothing to our manuscript.
The analytical parameters are also questionable; k = 1 implements clique analysis and should never be used in a parsimony setting; k <= 3 has been shown to typically produce less accurate trees. K = 10 is a widely recommended value and would be much more instructive. Please justify the selection of concavity constants used with reference to appropriate literature.
Reply: Repetitions with variable concavity values (k) were used to explore the effect of different degrees of homoplasy penalization to test the robustness of the dataset (Goloboff et al. 2008).
The text implies that all phylogenetic analyses place the new taxon in stem-Priapulida, but the equal weights results appear to show it in the crown group, with a crown group position also potentially included in the set of most parsimonious trees with k = 3 (depending on what topologies underpin the relevant polytomy).
Reply: We agree with you. We made the necessary corrections (see lines 248-253, 390-403).
Does the phylogenetic analysis not provide a basis for testing the authors’ statement that “the occurrence of '8 scalids around the first circle' seems not homologous between Priapulida and Loricifera.”? The text in any case contradicts this statement; line 295 reports “found exclusively in the crown-group Priapulida”. It is also necessary to discuss which Cambrian taxa this character can be scored unequivocally for, as it requires exceptional preservation to be able to distinguish and count this first circlet precisely. Is the state really known with confidence outside extant taxa?
Reply: As added to the text of our revised version, “8 scalids around the first circle” is considered as homologue for Loricifera and Priapulida, based on neuroanatomy (see lines 271-280), although it is not supported by current and published phylogenetic analyses (Wills et al. 2012, Sorensen et al. 2008, Laumer et al. 2019) that were based on morphological and molecular data.

Round 2
Reviewer 1 Report
I am pleased to see that the authors extensively revised the manuscript, in form and content, with entire sections completely reorganized. It is evident that the authors invest time and effort in preparing this new version, and the result is clearly satisfactory.
I think that the work is now acceptable for publication.
There is a minor edit that, in my opinion, can be done directly when revising the proof:
Line 341:
McMenamin (2016) to be formatted.
Author Response
We thank you for your constructive critical comments and revised our MS accordingly. All modifications in the revised version are marked in red. Numbers (e.g. L-01) used in the following point-by-point reply refer to the corresponding line numbers in the revised MS.
Best regards,
All authors
Line 341: McMenamin (2016) to be formatted.
Reply: OK, done.
Reviewer 4 Report
This paper is ready for publication as revised, after fixing these corrections to the text:
17 italics for sparios, followed by gen. et sp. nov.
342 change 'six laws' to 'nine laws'
419 add space before, then italics for Tubiluchus troglodytes
513 italics needed for taxon name
516 italics needed for taxon name
Author Response
We thank you for your constructive critical comments and revised our MS accordingly. All modifications in the revised version are marked in red. Numbers (e.g. L-01) used in the following point-by-point reply refer to the corresponding line numbers in the revised MS.
Best regards,
All authors
17 italics for sparios, followed by gen. et sp. nov.
Reply: OK, done, similar mistakes have been corrected thorough the manuscript, see revised version.
342 change 'six laws' to 'nine laws'
Reply: OK, done, similar mistakes have been corrected thorough the manuscript, see revised version.
419 add space before, then italics for Tubiluchus troglodytes
Reply: OK, done, similar mistakes have been corrected thorough the manuscript, see revised version.
513 italics needed for taxon name
Reply: OK, done, similar mistakes have been corrected thorough the manuscript, see revised version.
516 italics needed for taxon name
Reply: OK, done, similar mistakes have been corrected thorough the manuscript, see revised version.
Reviewer 5 Report
Although the authors have made some highly worthwhile improvements to the visualization of the fossil, they still do not conclusively demonstrate the presence of 16 (rather than a more parsimonious, if still novel, 15) longitudinal rows. The revised fig. 1 is helpful, but I still cannot distinguish lines 8, 9 and 10 with sufficient clarity to be confident that there are three rows here rather than two. Likewise there are a number of “rows” (e.g. L16) represented by a single scalid, where it is difficult to be confident of the precise layout. Meanwhile, a possible row between the authors’ L3 and L4, whose anteriormost sclerite is quite well defined, is unmarked and unnumbered. Whilst the majority of the description is sound and worthy of publication, the detailed arrangement of introvert scalids – and hence the central basis of the discussion – is at best tentative. Before this can be considered for publication, I would like to see a parallel consideration of the two key interpretations: namely, the text should also fully consider the implications of the possibility that there are 15 longitudinal rows (leaving a five-fold symmetry).
The objective of the study also needs to be articulated consistently and clearly: I think the intention is to define a set of morphological criteria by which fossil taxa can be assigned to the Priapulida crown, but the basis on which these criteria might be applied needs to be clearly articulated, and the suggestion that morphology can “define” the crown group must be removed; this is mostly a case of being careful in the use of terminology, but the present loose usage of language obscures what the authors are trying to accomplish.
Finally, the phylogenetic results that underpin the authors’ discussion are questionable; the authors rely on Fitch parsimony (which has been shown to produce artefacts on this particular dataset) and ignore the results of the Bayesian analysis that they perform, which produce a very different phylogenetic scenario.
Further, more detailed comments follow below.
Introduction. What is missing from the introduction is a clear statement to the effect of “even though a crown group is defined by phylogenetic relationships and not by a suite of morphological characters, we propose that the crown group is defined by features X, Y and Z based on (references to appropriate studies, or principled argument)”. Without this statement, later comments – e.g. that 8 scalids in the first circlet only occurs in the crown group – come out of nowhere, and feel like straw man arguments – has anyone ever assumed this, and if so, why?
The phylogenetic discussion is predicated on parsimony results which are likely to be misleading for the Shi et al. dataset, as shown by Smith & Dhungana 2022 Geol Mag – which should surely be discussed in the context of the phylogenetic results? The authors mention in their response document that BGS analysis results in less resolution: which is itself an important result, suggesting that the Fitch parsimony results offer false resolution and may not be reliable. It may be worth conducting rogue taxon analysis to see whether this polytomy could be resolved; but possibly the BGS results recover variants of both the Bayesian and Fitch trees as possible solutions?
Owing to this uncertainty, it’s not clear why the Bayesian results were not used as a basis for the discussion; at a minimum the authors should demonstrate that the parsimony results stand up when inapplicables are handled appropriately. Bayesian inference returns a profoundly different topology, which would have very different implications for many of the authors’ conclusions (as many taxa, e.g. Eximipriapulus, are not assigned to the Priapulid total group in these results).
The mathematical treatment of spine spacing is at best crude, and cannot be used to support an exact number of rows. The spacing can clearly be seen to decrease towards the margins (as expected due to compression). But it is not possible to reconstruct how compression occurred; the width of the introvert decreases anteriad; the simple measurements model the cross-section of the specimen as quadrilateral rather than elliptical; there is no allowance for rows potentially being preserved obliquely… and so forth. Perhaps the maths is adequate to be confident there weren’t 25 or 10 rows in total – but we could get this by simply doubling the number seen on a single surface. If the authors really insist on taking this approach, then they must quantify all possible sources of uncertainty and thus present their Nrm with a statistically valid estimation of error. I would be very surprised if this error envelope did not encompass both 15 and 16 as possible interpretations. Moreover, this treatment depends on the authors’ successful identification of all longitudinal rows; the additional unmarked row between L3 and L4 would change this calculation.
Line 15. “The problem lies in the morphological features (ornament, symmetry) currently used to define priapulids”: false; the problem lies in the lack of morphological detail available from Cambrian “worms” and the fact that a comprehensive phylogeny of Cambrian worms with a robust homology framework is not available.
Line 19. “Was selected to become” reads as if an external agent (an “intelligent designer”?) made the decision to select. Please reword to avoid this potential misreading.
Line 21. Very few Palaeozoic worms are resolved as crown-group priapulans.
L23. “That both create” reads as if to mean “both pharyngeal teeth… create symmetry patterns”. Please reword.
L25. A five-fold symmetry is implied by 25 longitudinal rows of scalids; please explain how these two features are distinct.
L27. Please remove the value judgement “more interestingly”. If an adjective is required here, “more speculatively” would be more appropriate.
L33. I can’t guess what is meant by “possibly via the standardization of patterning”.
L38. Is the monophyly of Scalidophora universally supported by all studies, as implied here?
L38. “Relatively small number” is vague; relative to what? Suggest specifying an approximate value (e.g. comprising just a few dozen extant species distributed…)
L61. Why “Over-influenced”? You are suggesting that the taxon does not belong to the crown group, but need to explain why. You have asserted that 25 scalid rows defines the crown, but not demonstrated that this was the case in the ancestor of the crown group, nor established that this number could not have been secondarily reduced in Priapulites. The implicit assumption that 8+9+8 is homologous to 8+8+9 is not stated or defended. The claim regarding Priapulites should be expanded to explain whether the caudal appendages characterize a subgroup of extant priapulids (in which case they may denote membership of that subgroup) or all the crown group (in which case they could also have occurred in some stem-group members – but consequently would be expected in all crown priapulans).
L64. It would be worth spelling out exactly what you mean by “symplesiomorphy” in this context: I read it to imply that the character evolved in the common ancestor of extant priapulids, and not a moment sooner or later (which is a straw man argument; clearly the odds of any single character happening to evolve at precisely this moment are close to zero), but it could also be taken to mean that it arose in the stem lineage (which would be consistent with its observation in e.g. Markuelia if this taxon was in fact a priapulan).
L72: “the same difficulties as scalids” presumably refers to preservational difficulties, but these have not been explicitly discussed (whereas other difficulties of scalids have). Please spell out exactly what you mean.
L249: Polytomies reduce precision, not accuracy. It’s not possible to establish whether the topoloiges are accurate without knowing the true relationships. I think what the authors mean to say here is that “low values of k, and equal weights, tend to be less accurate methods”. Why does this section not discuss ML/ Bayesian results?
L268-278. This is clear, easy to follow, and a good summary of the available evidence.
L279. This is a non-question: a crown-group Priapulida is defined as any taxon descended from the common ancestor of all extant Priapulida. I think what the authors mean is “how can we *recognize* a crown-group priapulid”.
L286-288. Priapulids inhabit a range of environments, some of which (e.g. meiofaunal settings) they did not inhabit in the Cambrian.
L286-288, “conservatism”, “virtually unchanged”: This is contrary to the results of Wills et al. 2012, which show that Cambrian and Recent priapulomorphs occupy non-overlapping regions of morphospace. The authors need to specify in the text that these quantitative results go against their thesis, and to state why they choose to overlook them.
L291-296. This description is incomplete and should be self-contained; do authors contend that any of the three characters is sufficient to diagnose a crown-priapulan, or that all three must be present in combination (which would make moot the statements that individual features can be found in isolation in other groups).
Line 311-314. It is confusing to list Cambrian taxa here, as these are not unequivocally members of the crown group – and indeed some are resolve in the stem group in the authors’ analysis.
Lines 331-332. Palaeoscolecids are (probably) not stem-priapulids. This is important, as it undermines the authors’ interpretation of “diverse” symmetry: if six-fold was present in the ancestral ecdysozoan, then five-fold symmetry may characterize a single clade containing priapulids and some relatives.
L347-351. There is a contradiction here: if a trait is under (natural) selection then it must confer a selective advantage. But if the number of scalid rows had a “very limited impact” it is inconsistent to argue that it should be under strong selection.
Fig. 5. The low-quality rendering of the image in the review PDF means that the image is almost illegible – hopefully the publisher will reproduce the figure better in the final version of the paper. As in the last version, the key does not match the symbols actually used; Ercaivermis looks like it is depiced with the cyan hexaradial symmetry marker, rather than grey octaradial.
Fig. A4. Total Priapulida is mis-labelled as it includes members of stem(Nematoida+Panarthropoda), including the palaeoscolecids as discussed above. Also this appears to be a majority rule (not strict) consensus tree; and the meaning of the (illegible) node supports should be explained in the caption.
See above
Author Response
Dear reviewers and editor,
We thank you for your constructive critical comments and revised our MS accordingly. All modifications in the revised version are marked in red. Numbers (e.g. L-01) used in the following point-by-point reply refer to the corresponding line numbers in the revised MS.
Best regards,
All authors
Reviewer 5
Although the authors have made some highly worthwhile improvements to the visualization of the fossil, they still do not conclusively demonstrate the presence of 16 (rather than a more parsimonious, if still novel, 15) longitudinal rows. The revised fig. 1 is helpful, but I still cannot distinguish lines 8, 9 and 10 with sufficient clarity to be confident that there are three rows here rather than two.
Reply: We try our best to make the number of longitudinal rows of scalids in Fig. 2 (see the revised Fig. 2) more clearly visible. The longitudinal rows labelled L7 to L10 (dashed lines in Fig. 2A-C). Scalids appear and are labelled on L7 and L10, respectively (in this revised version, Fig. 2A-C, E-G). L9 is recognized by the positive relief line (i.e. ridge see in Fig. 2B, C, F, G). Only one scalid (C4) of L9 can be ascertained in Fig.1D, F. L8 and its scalids are clearly shown in Fig. 1D.
Likewise there are a number of “rows” (e.g. L16) represented by a single scalid, where it is difficult to be confident of the precise layout.
Reply: Yes, it is marked one scalid on L16 in Fig. 1E, but another scalid of L16 is clearly shown in Fig. 1G. The other scalids e.g. C7 in L12, C6 in L11, are plotted on the Fig. 1H (see revised version).
Meanwhile, a possible row between the authors’ L3 and L4, whose anteriormost sclerite is quite well defined, is unmarked and unnumbered.
Reply: We see no sign of a possible row between the L3 and L4 except some small black concaves arrange irregularly. We also do not recognize the so-called “anteriormost sclerite” between the L3 and L4.
Whilst the majority of the description is sound and worthy of publication, the detailed arrangement of introvert scalids – and hence the central basis of the discussion – is at best tentative. Before this can be considered for publication, I would like to see a parallel consideration of the two key interpretations: namely, the text should also fully consider the implications of the possibility that there are 15 longitudinal rows (leaving a five-fold symmetry).
Reply: It may also be some other implications if Ercaivermis bears 15 longitudinal rows of scalids. However, the revised figures 1 and 2 are to us clear enough to attest for the presence of 16 longitudinal rows of scalids. Discussing the 15-rows-hypothesis in more details sounds unnecessary to us.
The objective of the study also needs to be articulated consistently and clearly: I think the intention is to define a set of morphological criteria by which fossil taxa can be assigned to the Priapulida crown, but the basis on which these criteria might be applied needs to be clearly articulated, and the suggestion that morphology can “define” the crown group must be removed; this is mostly a case of being careful in the use of terminology, but the present loose usage of language obscures what the authors are trying to accomplish.
Reply: Thanks for the helpful suggestions, we articulate relative content in the revised version despite defining a set of morphological criteria is not the key intention of this study. We also remove the “define” and relative loose usage of terminology.
Finally, the phylogenetic results that underpin the authors’ discussion are questionable; the authors rely on Fitch parsimony (which has been shown to produce artefacts on this particular dataset) and ignore the results of the Bayesian analysis that they perform, which produce a very different phylogenetic scenario.
Reply: Thanks for your suggestion. The results obtained via the Bayesian method are very uncertain due to the low values of posterior possibilities at key nodes (e.g. crown Priapulida (0.1), Nematoida+Panarthropoda (0.22), Palaeoscolecida+ (Nematoida+Panarthropoda) (0.06)). Parsimony results vary depending on the different values of k. In contrast, the bootstrap values are high (general above 50) in a result obtained via maximum likelihood to which we prefer to stick. See lines 242-256 in the revised version.
Further, more detailed comments follow below.
Introduction. What is missing from the introduction is a clear statement to the effect of “even though a crown group is defined by phylogenetic relationships and not by a suite of morphological characters, we propose that the crown group is defined by features X, Y and Z based on (references to appropriate studies, or principled argument)”. Without this statement, later comments – e.g. that 8 scalids in the first circlet only occurs in the crown group – come out of nowhere, and feel like straw man arguments – has anyone ever assumed this, and if so, why?
Reply: Thanks for your suggestion. We have revised it, see lines 52-56 in the revised version.
The phylogenetic discussion is predicated on parsimony results which are likely to be misleading for the Shi et al. dataset, as shown by Smith & Dhungana 2022 Geol Mag – which should surely be discussed in the context of the phylogenetic results? The authors mention in their response document that BGS analysis results in less resolution: which is itself an important result, suggesting that the Fitch parsimony results offer false resolution and may not be reliable. It may be worth conducting rogue taxon analysis to see whether this polytomy could be resolved; but possibly the BGS results recover variants of both the Bayesian and Fitch trees as possible solutions?
Reply: Thanks for your suggestion. We have revised it, see lines 242-256 in the revised version. See explanations above.
Owing to this uncertainty, it’s not clear why the Bayesian results were not used as a basis for the discussion; at a minimum the authors should demonstrate that the parsimony results stand up when inapplicables are handled appropriately. Bayesian inference returns a profoundly different topology, which would have very different implications for many of the authors’ conclusions (as many taxa, e.g. Eximipriapulus, are not assigned to the Priapulid total group in these results).
Reply: We ignore the results of the Bayesian analysis due to its low values of posterior possibilities on the key nodes, e.g. crown Priapulida (0.1), Nematoida+Panarthropoda (0.22), Palaeoscolecida+ (Nematoida+Panarthropoda) (0.06). The nodes of many ecdysozoan worms resolved as stem Nematoida+Panarthropoda are all below 0.1 (Fig. A4). This configuration is largely due to the large amount of missing and inapplicable characters in fossil taxa, an avoidable problem in palaeontology. Perhaps, BGS method knows how to treat inapplicable characters correctly. However, the results via BGS differ remarkably from the mainstream topologies of Ecdysozoa obtained from both molecular and morphological datasets. See lines 242-246 in the revised version.
The mathematical treatment of spine spacing is at best crude, and cannot be used to support an exact number of rows. The spacing can clearly be seen to decrease towards the margins (as expected due to compression). But it is not possible to reconstruct how compression occurred; the width of the introvert decreases anteriad; the simple measurements model the cross-section of the specimen as quadrilateral rather than elliptical; there is no allowance for rows potentially being preserved obliquely… and so forth. Perhaps the maths is adequate to be confident there weren’t 25 or 10 rows in total – but we could get this by simply doubling the number seen on a single surface. If the authors really insist on taking this approach, then they must quantify all possible sources of uncertainty and thus present their Nrm with a statistically valid estimation of error. I would be very surprised if this error envelope did not encompass both 15 and 16 as possible interpretations. Moreover, this treatment depends on the authors’ successful identification of all longitudinal rows; the additional unmarked row between L3 and L4 would change this calculation.
Reply: Thanks for your suggestion. We remove the table and measurements and mathematical calculations. Calculations are unnecessary here since CT-images show the presence of 16 scalid rows (see the revised Figs. 1, 2).
Line 15. “The problem lies in the morphological features (ornament, symmetry) currently used to define priapulids”: false; the problem lies in the lack of morphological detail available from Cambrian “worms” and the fact that a comprehensive phylogeny of Cambrian worms with a robust homology framework is not available.
Reply: Thank you. We have revised it, see lines 15-17 in the revised version.
Line 19. “Was selected to become” reads as if an external agent (an “intelligent designer”?) made the decision to select. Please reword to avoid this potential misreading.
Reply: Thank you. We have revised it, see line 20 in the revised version.
Line 21. Very few Palaeozoic worms are resolved as crown-group priapulans.
Reply: Thank you. We have revised it, see lines 22-23 in the revised version.
L23. “That both create” reads as if to mean “both pharyngeal teeth… create symmetry patterns”. Please reword.
Reply: Thank you. We have revised it, see lines 25-26 in the revised version.
L25. A five-fold symmetry is implied by 25 longitudinal rows of scalids; please explain how these two features are distinct.
Reply: Thank you. We have revised it, see lines 27-28 in the revised version.
L27. Please remove the value judgement “more interestingly”. If an adjective is required here, “more speculatively” would be more appropriate.
Reply: OK, done, the words are removed.
L33. I can’t guess what is meant by “possibly via the standardization of patterning”.
Reply: Thank you. We have revised it, see lines 35-36 in the revised version.
L38. Is the monophyly of Scalidophora universally supported by all studies, as implied here?
Reply: No, at least many molecular results do not support. The monophyly of Scalidophora was mostly resolved by the morphological data. We add words to clarify it, see revised version.
L38. “Relatively small number” is vague; relative to what? Suggest specifying an approximate value (e.g. comprising just a few dozen extant species distributed…)
Reply: Thank you. We have revised it, see line 41 in the revised version.
L61. Why “Over-influenced”? You are suggesting that the taxon does not belong to the crown group, but need to explain why. You have asserted that 25 scalid rows defines the crown, but not demonstrated that this was the case in the ancestor of the crown group, nor established that this number could not have been secondarily reduced in Priapulites.
Reply: Thank you. We have revised it, see line 68 in the revised version.
The implicit assumption that 8+9+8 is homologous to 8+8+9 is not stated or defended.
Reply: 8+8+9 pattern considered that 8 scalids of first circle differs from others because they are innervated by 8 remarkable cluster. And 8 scalids of the first circle do not place in 25 longitudinal rows of scalids defined by the following scalids. This is not correct (Schmidt-Rhaesa 2013, Handbook of Zoology. Vol. 2). Instead, 8+9+8 pattern only considers the number of scalids in each circle, it does not refer to the nerve aspects. The two patterns represent two different plans to define the 25 scalid rows, in fact, there are not substantially differences of both. (8+9+8) pattern is a widely accepted plan that we keep in manuscript. See lines 61-62 in the revised version.
The claim regarding Priapulites should be expanded to explain whether the caudal appendages characterize a subgroup of extant priapulids (in which case they may denote membership of that subgroup) or all the crown group (in which case they could also have occurred in some stem-group members – but consequently would be expected in all crown priapulans).
Reply: Thank you for the suggestions. Caudal appendages in extant priapulids occur in Priapulomorpha (Priapulidae and Tubiluchidae) but not all priapulids. Fossil Xiaoheiqingella, Yunnanpriapulus, Paratubiluchus, and Priapulites develop caudal appendages. These fossils were never consistently resolved as stem or crown priapulomorphs, sometimes they were resolved as stem priapulids (Wills et al. 2012, Harvey et al. 2010, Ma et al. 2014, Shi et al. 2021, Smith and Dhungana 2022). It remains unclear whether caudal appendages could be as a criterion for diagnosis of Priapulomorpha or Priapulida.
In the revised version, we think that “the presence of caudal appendages and 5-fold symmetry arrangement of scalid rows (20 in Priapulites)” may together influence to assign Priapulites to the crown priapulid. See lines 68-69 in the revised version.
L64. It would be worth spelling out exactly what you mean by “symplesiomorphy” in this context: I read it to imply that the character evolved in the common ancestor of extant priapulids, and not a moment sooner or later (which is a straw man argument; clearly the odds of any single character happening to evolve at precisely this moment are close to zero), but it could also be taken to mean that it arose in the stem lineage (which would be consistent with its observation in e.g. Markuelia if this taxon was in fact a priapulan).
Reply: It may be a mistake. “8 scalids around the first introvert circle” is a symplesiomorphy of the ground pattern of the Priapulida, see line 72 in the revised version.
L72: “the same difficulties as scalids” presumably refers to preservational difficulties, but these have not been explicitly discussed (whereas other difficulties of scalids have). Please spell out exactly what you mean.
Reply: yes, the compression of fossils can lead to difficultly recognize the exact number of teeth of circles. In addition, inverted pharynx also obscures to count the number of teeth e.g. in Ercaivermis. This has been explained in the revised version, see lines 79-81.
L249: Polytomies reduce precision, not accuracy. It’s not possible to establish whether the topoloiges are accurate without knowing the true relationships. I think what the authors mean to say here is that “low values of k, and equal weights, tend to be less accurate methods”. Why does this section not discuss ML/ Bayesian results?
Reply: Thank you for the suggestions. Please see the revised paragraph (lines 242-256) to discuss the ML and Bayesian results.
L268-278. This is clear, easy to follow, and a good summary of the available evidence.
Reply: Thanks.
L279. This is a non-question: a crown-group Priapulida is defined as any taxon descended from the common ancestor of all extant Priapulida. I think what the authors mean is “how can we *recognize* a crown-group priapulid”.
Reply: OK, done, see line 286 in the revised version.
L286-288. Priapulids inhabit a range of environments, some of which (e.g. meiofaunal settings) they did not inhabit in the Cambrian.
Reply: Thank you for the suggestions. Perhaps, the micro-meter-sized stem-priapulids from the Kuanchuanpu biota may be meiofaunal species, e.g. Xinliscolex (Zhang H.Q. 2021, Palaeoworld), Qinscolex, Shanscoelx, and other indeterminate forms (Liu Y.H. et al. 2018, Papers in Palaeontology).
L286-288, “conservatism”, “virtually unchanged”: This is contrary to the results of Wills et al. 2012, which show that Cambrian and Recent priapulomorphs occupy non-overlapping regions of morphospace. The authors need to specify in the text that these quantitative results go against their thesis, and to state why they choose to overlook them.
Reply: Thank you for the suggestions. The “conservatism” and ”virtually unchanged” focus on the body plan of priapulids, i.e. three-fold introvert, annulated trunk, and caudal appendages. Variations mainly concern the radial symmetry of scalid rows, the number of longitudinal rows of scalids on the introvert, number and arrangement of sclerites on the trunk, and so on, as Wills et al. 2012 suggested. See lines 293-296, 374 in the revised version.
L291-296. This description is incomplete and should be self-contained; do authors contend that any of the three characters is sufficient to diagnose a crown-priapulan, or that all three must be present in combination (which would make moot the statements that individual features can be found in isolation in other groups).
Reply: Morpho-anatomy aspects, the diagnoses of extant priapulids are not limited to these three features mentioned here. See explanations in lines 309-310 in the revised version.
Line 311-314. It is confusing to list Cambrian taxa here, as these are not unequivocally members of the crown group – and indeed some are resolve in the stem group in the authors’ analysis.
Reply: Thank you for the suggestions. We remove these fossil examples in the text.
Lines 331-332. Palaeoscolecids are (probably) not stem-priapulids. This is important, as it undermines the authors’ interpretation of “diverse” symmetry: if six-fold was present in the ancestral ecdysozoan, then five-fold symmetry may characterize a single clade containing priapulids and some relatives.
Reply: It depends on whether palaeoscolecids are truly stem-priapulids or not. It remains open debate.
L347-351. There is a contradiction here: if a trait is under (natural) selection then it must confer a selective advantage. But if the number of scalid rows had a “very limited impact” it is inconsistent to argue that it should be under strong selection.
Reply: Thank you for the suggestions. We correct in the revised version, see lines 20, 35, 349, 352, 355.
Fig. 5. The low-quality rendering of the image in the review PDF means that the image is almost illegible – hopefully the publisher will reproduce the figure better in the final version of the paper. As in the last version, the key does not match the symbols actually used; Ercaivermis looks like it is depiced with the cyan hexaradial symmetry marker, rather than grey octaradial.
Reply: The mistakes have been already corrected (see revised version). The low-quality resolution of Fig. 5 in the review is due to a compressed version. If the manuscript is accepted, then we will provide a high-quality file of Fig. 5.
Fig. A4. Total Priapulida is mis-labelled as it includes members of stem(Nematoida+Panarthropoda), including the palaeoscolecids as discussed above. Also this appears to be a majority rule (not strict) consensus tree; and the meaning of the (illegible) node supports should be explained in the caption.
Reply: Thank you for the suggestions. We make the corrections on Fig. A4, please see the revised version.

Round 3
Reviewer 5 Report
I thank the authors for their careful work addressing the previous comments.
However, despite the tweaks to the figures, I am still unconvinced by the authors’ interpretation of 16 longitudinal rows (and thus eight-fold symmetry). The bottom line here is that the single available specimen does not provide the unambiguous depiction of morphology necessary to support this claim; in the presence of imperfect preservation and taphonomic deformation, more specimens are necessary. As such, I cannot support the publication of the paper unless this claim is presented as one of many possible reconstructions of the fossil.
A couple of small points arising from the authors’ changes:
Fig. 2 worked better with minimal overlay of annotations on the second row, such that the full fossil anatomy can be reflected without obstruction; the first row suffices to label features.
Lines 248–256. The wording portrays a lack of resolution (i.e. uncertainty) as a “drawback” of two methods. On the contrary, if this uncertainty is a genuine reflection of uncertainty in the data, then this is a *strength* of the methods: the methods do not display over-confidence in a scenario that is likely to be incorrect, but identifies areas of the tree that are uncertain based on the data (likely due to the uncertain placement of a small number of “rogue” taxa).
Line 309. Despite the changes made above this repeats the misconception that the crown-group is defined on the basis of morphology. Suggest “All these factors make it difficult to identify a crown-group priapulid based on morphological features alone”.
Line 372. If the body plan of crown-priapulids is difficult to define, then it doesn’t follow that the body plan has remained unchanged for 500 Ma; in fact this (alongside examples of divergence such as Meiopriapulius) suggests the opposite.
Fig. A4 caption. Note misspelling of “majority”.
n/a
Author Response
Dear reviewers and editor,
We thank you for your constructive critical comments and revised our MS accordingly. All modifications in the revised version are marked in red. Numbers (e.g. L-01) used in the following point-by-point reply refer to the corresponding line numbers in the revised MS.
Best regards,
All authors
Reviewer 5
I thank the authors for their careful work addressing the previous comments.
However, despite the tweaks to the figures, I am still unconvinced by the authors’ interpretation of 16 longitudinal rows (and thus eight-fold symmetry). The bottom line here is that the single available specimen does not provide the unambiguous depiction of morphology necessary to support this claim; in the presence of imperfect preservation and taphonomic deformation, more specimens are necessary. As such, I cannot support the publication of the paper unless this claim is presented as one of many possible reconstructions of the fossil.
Reply: we thank you for the suggestion. Based on the single available specimen, we described the arrangement of scalid rows as accurately as possible (Figures 1D, G, H, 2). It seems that we failed convincing you that, to us, 16 rows of scalids are most likely present. Unfortunately, we don't think it's worth continuing endless discussions on this topic.
In addition, we add the number of longitudinal rows of scalids at some taxa in the Figure 5 (see number in the paired brackets). Due to one radial symmetry could result from different number of longitudinal rows of scalids (e.g. 5-fold symmetry can result from 20 or 25 scalid rows). It is necessary to make it clear.
A couple of small points arising from the authors’ changes:
Fig. 2 worked better with minimal overlay of annotations on the second row, such that the full fossil anatomy can be reflected without obstruction; the first row suffices to label features.
Reply: OK, done.
Lines 248–256. The wording portrays a lack of resolution (i.e. uncertainty) as a “drawback” of two methods. On the contrary, if this uncertainty is a genuine reflection of uncertainty in the data, then this is a *strength* of the methods: the methods do not display over-confidence in a scenario that is likely to be incorrect, but identifies areas of the tree that are uncertain based on the data (likely due to the uncertain placement of a small number of “rogue” taxa).
Reply: Thank you for the suggestions. In all methods, Ercaivermis, is consistently resolved as a stem priapulid that is the closest taxon to the crown priapulids. It at least suggests that the placement of Ercaivermis is reliable, despite the other part of the Bayesian and parsimony topologies may be contentious.
Line 309. Despite the changes made above this repeats the misconception that the crown-group is defined on the basis of morphology. Suggest “All these factors make it difficult to identify a crown-group priapulid based on morphological features alone”.
Reply: Thank you for the suggestions. OK, done.
Line 372. If the body plan of crown-priapulids is difficult to define, then it doesn’t follow that the body plan has remained unchanged for 500 Ma; in fact this (alongside examples of divergence such as Meiopriapulius) suggests the opposite.
Reply: Thank you for the suggestions. The body plan of Priapulida consists of pharynx, introvert, trunk, and neck, caudal appendages (if present). If we compare an extant species such as Priapulus with a Cambrian one (regardless of their placement in the stem- or crown-group), we see that the body plan of these worms has virtually remained unchanged for 500 Ma. However, differences and variations can be found at other anatomical levels such as the distribution and number of scalid rows (20, 25, >30), symmetry of pharyngeal teeth (5, 8).
Meiofaunal representatives of the group evolved towards miniaturization and show morphological adaptations to their lifestyle and habitat. However, they have the same overall body plan as their macrobenthic counterparts and Cambrian forms. We are aware that defining crown priapulids is not an easy task. Using characters that define the overall body plan of priapulids would not be a good choice. Instead, we suggest to consider other morphological characters. We retain the first half of the sentence and remove the second half of it. See the revised version.
Fig. A4 caption. Note misspelling of “majority”.
Reply: Thank you for the suggestions. OK, done.